# IT³: IDEMPOTENT TEST-TIME TRAINING

## ABSTRACT

This paper introduces *Idempotent Test-Time Training* (IT³), a novel approach to addressing the challenge of distribution shift. While supervised-learning methods assume matching train and test distributions, this is rarely the case for machine learning systems deployed in the real world. Test-Time Training (TTT) approaches address this by adapting models during inference, but they are limited by a domain specific auxiliary task. IT³ is based on the universal property of idempotence. An idempotent operator is one that can be applied sequentially without changing the result beyond the initial application, that is $f(f(\mathbf{x})) = f(\mathbf{x})$. At training, the model receives an input $\mathbf{x}$ along with another signal that can either be the ground truth label $\mathbf{y}$ or a neutral "don't know" signal $\mathbf{0}$. At test time, the additional signal can only be $\mathbf{0}$. When sequentially applying the model, first predicting $\mathbf{y}_0 = f(\mathbf{x}, \mathbf{0})$ and then $\mathbf{y}_1 = f(\mathbf{x}, \mathbf{y}_0)$, the distance between $\mathbf{y}_0$ and $\mathbf{y}_1$ measures certainty and indicates out-of-distribution input $\mathbf{x}$ if high. We use this distance, that can be expressed as $||f(\mathbf{x}, f(\mathbf{x}, \mathbf{0})) - f(x, \mathbf{0})||$ as our TTT loss during inference. By carefully optimizing this objective, we effectively train $f(\mathbf{x}, \cdot)$ to be idempotent, projecting the internal representation of the input onto the training distribution. We demonstrate the versatility of our approach across various tasks, including corrupted image classification, aerodynamic predictions, tabular data with missing information, age prediction from face, and large-scale aerial photo segmentation. Moreover, these tasks span different architectures such as MLPs, CNNs, and GNNs.

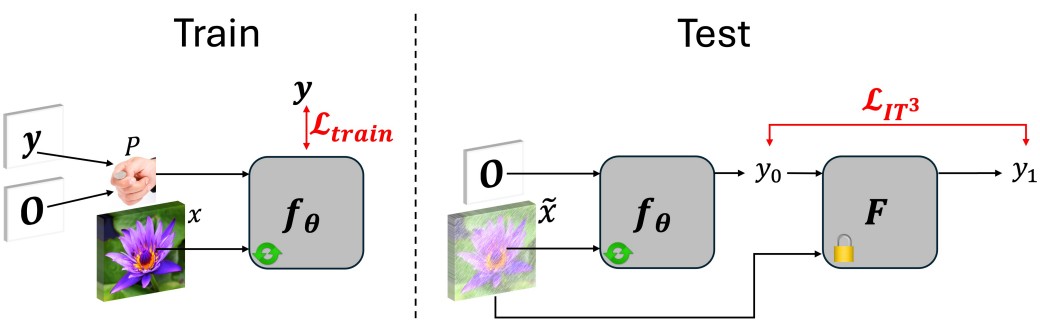

Figure 1: **Idempotent Test-Time Training (IT³) approach.** During training (left), the model $f_\theta$ is trained to predict the label $y$ with or without $y$ given to it as input. At test time (right), when given a corrupted input, the model is sequentially applied. It then briefly trains with the objective of making $f_\theta(\mathbf{x}, \cdot)$ to be idempotent using only the current test input.

## 1 INTRODUCTION

Supervised learning methods, while powerful, typically assume that training and test data come from the same distribution. Unfortunately, this is rarely true in practice. Data encountered by systems operating in the real world often differs substantially from what they were trained on due to data distribution shifts over time or other changes in the environment. This inevitably degrades performance, even in state-of-the-art models (Recht et al., 2018; Hendrycks et al., 2021; Yao et al.,

2022). Modern deployed machine learning systems not only need to adapt to distribution shifts but also must do so on-the-fly using very limited data.

The problem setup this work addresses requires adapting to distribution shifts on-the-fly using only the current test instance or batch, without access to any additional labeled or unlabeled data during inference. During training, the model has access solely to the base distribution training data, with no prior knowledge of the test distribution. Adversarial robustness and domain adaptation address related challenges, they typically require additional data either during training or inference, and sometimes rely on specific assumptions about the nature of the shift. While effective in their contexts, they are not designed for immediate, instance-level adaptation and do not solve our problem setup. Test-Time Training (TTT)(Sun et al., 2020) offers an alternative by adapting the model during inference using an auxiliary self-supervised task on each test sample. This dynamic, on-the-fly adaptation allows the model to handle corrupted and Out-of-Distribution (OOD) data using only the current test instance or batch, without access to any other data. However, TTT employs an auxiliary task specific to the data modality (e.g., orientation prediction or inpainting for imagery data)(Gandelsman et al., 2022).

In this paper, we argue that enforcing *idempotence* can profitably replace the auxiliary tasks in TTT and results in an approach we dub IT$^3$ that is a versatile and powerful while generalizing well across domains and architectures.

More specifically, an operator $f$ is said to be idempotent if it can be applied sequentially without changing the result beyond the initial application, namely: $f(f(\mathbf{x})) = f(\mathbf{x})$. This can be understood as a generalization of orthogonal projection in linear spaces to non-linear settings. At training, IT$^3$ receives an input $\mathbf{x}$ along with another signal that can either be the ground truth label $y$ or a neutral "don't know" signal $\mathbf{0}$. Durasov et al. (2024a) Sequentially applied a model that was trained with this policy s.t. $y_0 = f(\mathbf{x}, \mathbf{0})$ and $y_1 = f(\mathbf{x}, y_0)$. The distance $||y1 - y2||$ in some metric, indicates the prediction uncertainty and also indicates whether $\mathbf{x}$ is OOD. What if, at test time, we could actively minimize this distance whenever we encounter an instance? Could we "pull it" into the distribution? IT$^3$ uses this this distance as a loss for TTT sessions. When we unfold $y_1$ and $y_2$ in such a loss term we obtain: $||f(\mathbf{x}, f(\mathbf{x}, \mathbf{0})) - f(\mathbf{x}, \mathbf{0})||$. Closer examination of this term reveals a key insight: the optimization objective is actually driving the model to make $f(\mathbf{x}, \cdot)$ *Idempotent*! While not trivial, we know that, with careful adjustments, it is indeed possible to train a model to be idempotent (Shocher et al., 2024). This ties everything together: idempotence, seen as a generalization of projection, suggests the existence of a subset onto which the model maps the internal representation of th input. In our case, this subset exists in the joint $X \times Y$ space, and corresponds roughly to the distribution of correctly paired $\mathbf{x}, \mathbf{y}$ examples.

The result is a global method that does not rely on any specific domain properties. This is in contrast to prior TTT methods that rely on a domain specific auxiliary task. By leveraging the universal property of idempotence, IT$^3$ can adapt OOD test inputs on-the-fly across various domains, tasks and architectures. This includes image classification with corruptions, aerodynamic predictions for airfoils and cars, tabular data with missing information, age prediction from faces, and large-scale aerial photo segmentation, Using MLPs, CNNs or GNNs.

## 2 RELATED WORK

IT$^3$ relies on the notion of *idempotence* to globalize *TTT*. We briefly review these two fields.

### 2.1 TEST-TIME TRAINING

The idea of leveraging test data for model adaptation dates back to methods like transductive learning (Gammerman et al., 1998). Early approaches, such as transductive SVMs (Collobert et al., 2006) and local learning (Bottou & Vapnik, 1992), aimed to adapt predictions for specific test samples rather than generalizing across unseen data.

Training neural networks solely on single test instances, without pre-training, has been demonstrated in the "deep internal learning" line of works, for many image enhancement tasks (Shocher et al., 2018; Gandelsman et al., 2019) and single image generative models (Shocher et al., 2019; Shaham et al., 2019).

Test-Time Training (TTT) has emerged as a solution to the problem of generalization under distribution shifts. Using a pre-trained network and at test-time refining on a single instance each time. In the foundational work of Sun et al. (2020), the model is adjusted in real-time by solving an auxiliary self-supervised task, such as predicting image rotations, on each test sample. This on-the-fly adaptation has proven effective in improving robustness on corrupted and Out-Of-Distribution (OOD) data. As the self-supervised learning methods became more efficient (He et al., 2022), they could be exploited for TTT (Gandelsman et al., 2022). Extensions such as TTT++ (Liu et al., 2021) assume access to the entire test set. TENT (Wang et al., 2021) adapts during inference in the batch level, based on the batch entropy, but cannot be applied to single instances or very small batches. Moreover, it relies on updating the model's normalization layers, making it architecture dependent.

## 2.2 IDEMPOTENCE IN DEEP LEARNING

Idempotence, a concept rooted in mathematics and functional programming, refers to an operation where repeated application yields the same result as a single application. Mathematically, for a function $f$, being idempotent means

$$f(f(x)) = f(x), \quad \forall x .\tag{1}$$

In other words, applying the function multiple times has no effect beyond the first application. In the realm of linear operators, idempotence equates to orthogonal projection. Over $\mathbb{R}^n$, these are matrices $A$ that satisfy $A^2 = A$, with eigenvalues that are either 0 or 1; they can be interpreted as geometrically preserving certain components while nullifying others. This principle was recently used for generative modeling. Idempotent Generative Network (IGN) (Shocher et al., 2024) is a generative model based on mapping data instances to themselves $f(x) = x$ and map latents to targets that map to them selves $f(f(z) = f(z)$. It was further shown to be able to 'project' corrupted images onto the data manifold, practically remove the corruptions with no prior knowledge of the degradation .

Energy-Based Models (EBMs; Ackley et al. (1985)) offer a related perspective by defining a function $f$ that assigns energy scores to inputs, with higher energy indicating less desirable or likely examples, and lower energy indicating those that fit the model well. IGN introduces a similar concept but frames it differently: instead of $f$ directly serving as the energy function, the energy is implicitly defined via the difference $\delta(y) = D(f(y), y)$, where $D$ measures the distance between the model's prediction and its input. In this framework, training $f$ to be idempotent minimizes $\delta(f(z))$, pushing the model toward a low-energy configuration where its outputs remain stable under repeated applications. Thus, $f$ can be interpreted as a transition operator that drives high-energy inputs toward a low-energy, stable domain, reducing the need for separate optimization procedures to find the energy minimum.

In concurrent work, the ZigZag method has first been proposed and then extended to recursive networks (Durasov et al., 2024b;a). It introduces idempotence as a means to assess uncertainty in neural network predictions, ZigZag operates by recursively feeding the model's predictions back as inputs, allowing the model to refine its outputs. The consistency between successive predictions acts as an uncertainty metric, where stable, unchanged outputs indicate higher confidence, while divergent predictions signal uncertainty or out-of-distribution (OOD) data. Unlike popular sampling-based uncertainty estimation methods (Gal & Ghahramani, 2016; Lakshminarayanan et al., 2017; Wen et al., 2020; Durasov et al., 2021), ZigZag does not require many forward passes or complex sampling, making it more computationally efficient for real-time applications.

## 3 METHOD

### 3.1 INITIAL TRAINING

Let $f_\theta$ be a generic model that we wish to deploy in an environment where the statistical distribution of the samples it receives may change over time. Fig 1 depicts the initial training phase, we perform a standard supervised training with a slight modification inspired by the ZigZag method Durasov et al. (2024b): We modify the first layer of the network implementing $f_\theta$ so that it can accept a second argument in addition to the data sample $\mathbf{x}$ that it is normally takes as input. This additional argument can be either $\mathbf{y}$, the desired output of the network given input $\mathbf{x}$, or a neutral "don't know"

signal $\mathbf{0}$. During training, we minimize the supervised loss

$$\mathcal{L}_{\text{train}} = \|f_\theta(\mathbf{x}, \mathbf{y}) - \mathbf{y}\| + \|f_\theta(\mathbf{x}, \mathbf{0}) - \mathbf{y}\| , \tag{2}$$

where $f(\mathbf{x}, \cdot)$ is the model's prediction given input $\mathbf{x}$ and the additional input. when $\mathcal{L}_{\text{train}}$ is minimized, we can write

$$\mathbf{y}_0 = f_\theta(\mathbf{x}, \mathbf{0}) \approx \mathbf{y} , \mathbf{y}_1 = f_\theta(\mathbf{x}, \mathbf{y}_0) \approx f_\theta(\mathbf{x}, \mathbf{y}) \approx \mathbf{y} \Rightarrow f_\theta(\mathbf{x}, f_\theta(\mathbf{x}, \mathbf{0})) \approx f_\theta(\mathbf{x}, \mathbf{0}) . \tag{3}$$

Of course, this can only be expected to hold when $\mathbf{x}$ is within the training distribution. When $\mathbf{x}$ is out-of-distribution, $\mathbf{y}_0$ and $\mathbf{y}_1$ can be very different. In Durasov et al. (2024b), this is exploited to estimate uncertainty: the greater the deviation from Eq. 3, the more uncertain the prediction is taken to be. In contrast, in this paper, we use Test-Time Training to enforce the constraint of Eq. 3 during inference.

## 3.2 TEST-TIME TRAINING

At test time, IT$^3$ introduces a dynamic adaptation process to refine the model's predictions for new inputs, particularly those that may be OOD or corrupted. The goal is to make the model idempotent with respect to its second input, that is, ensuring that $f_\theta(\mathbf{x}, \cdot)$ is an idempotent function for a given $\mathbf{x}$. A naive way to enforce this would be to minimize the loss function

$$\mathcal{L}_{\text{TTT}} = \|f_\theta(\mathbf{x}, f_\theta(\mathbf{x}, \mathbf{0})) - f_\theta(\mathbf{x}, \mathbf{0})\| . \tag{4}$$

However, directly minimizing this loss can produce undesirable side effects. For instance, if $\mathbf{y}_0$ is an incorrect prediction, minimizing the distance $\|\mathbf{y}_0 - \mathbf{y}_1\|$ may cause $\mathbf{y}_1$ to be pulled toward the incorrect $\mathbf{y}_0$, thereby magnifying the error. Another potential failure mode is due to the fact that *identity* is idempotent. For $f = identity$, we get $\mathcal{L}_{\text{TTT}} = 0$.

To prevent such a collapse, we modify the test-time training procedure as depicted in Fig, 1: We keep a copy of the model as it was at the end of the training phase, denoted as $F = f_\Theta$, where $\Theta$ are the weights obtained after the initial training of Sec. 3.1, which will not be updated further. We then take the test-time loss to be

$$\mathcal{L}_{\text{IT}^3} = \|F(\mathbf{x}, f_\theta(\mathbf{x}, \mathbf{0})) - f_\theta(\mathbf{x}, \mathbf{0})\| , \tag{5}$$

where $f_\theta$ is the model being updated at test-time. Here, the first prediction $\mathbf{y}_0 = f_\theta(\mathbf{x}, \mathbf{0})$ is computed as before, but the second one, $\mathbf{y}_1 = F(\mathbf{x}, f_\theta(\mathbf{x}, \mathbf{0}))$, is made using the frozen model $F$. By updating only $f_\theta$ and keeping $F$ fixed, we ensure that $\mathbf{y}_0$ is adjusted to minimize the discrepancy with $\mathbf{y}_1$, without pulling $\mathbf{y}_1$ toward an incorrect $\mathbf{y}_0$. A similar idea was employed in the IGN approach (Shocher et al., 2024) meaningful predictions are required. After each TTT optimization iteration, the dynamic model $f_\theta$ is initialized with $\Theta$, ready for the next input.

## 3.3 ONLINE IT$^3$

For streaming data scenarios, where the distribution shifts continuously over time, we modify IT$^3$ to operate in an online mode by not resetting $f_\theta$ back to $F$ after each TTT episode, as we did in Section 3.2. We essentially assume that the distribution mostly shifts smoothly and, thus, there is a good reason to believe that the current state of $f_\theta$ is a better initialization for the next TTT episode than the original $F$. This makes the model evolve over time. In this scheme, it can happen that the performance of the model on data from its original training decreases significantly, a phenomenon known as catastrophic forgetting (Kirkpatrick et al., 2017). This is acceptable as the goal is to perform well on data at the present moment, rather than on past examples.

Another modification in the online setup is for the second sequential application of the model, i.e., the $F$ in $F(\mathbf{x}, f_\theta(\mathbf{x}, \mathbf{0}))$. Since the data keeps shifting, there is no reason to retain the frozen $F$ as an anchor indefinitely. Over time, $f_\theta$ may diverge far from $F$, making it irrelevant. Relying on the old state of the model would prevent the model from evolving efficiently. Replacing it with the current state of $f_\theta$ is out of the question, as it causes collapse. We need an anchor that is influenced by a reasonable amount of data, yet evolves over time. Our solution is to replace $F$ with an Exponential Moving Average (EMA) of the model $f_\theta$, denoted as $f_{\text{EMA}}$. This means $f_{\text{EMA}}$ is a smoothed version of $f_\theta$ over time. The test-time loss in the online setting then becomes

$$\mathcal{L}_{\text{online}} = \|f_{\text{EMA}}(\mathbf{x}, f_\theta(\mathbf{x}, \mathbf{0})) - f_\theta(\mathbf{x}, \mathbf{0})\| . \tag{6}$$

By updating both $f_\theta$ and $f_{\text{EMA}}$ incrementally, with $f_{\text{EMA}}$ serving as a stable reference that changes more slowly, the model adapts to gradual shifts without overfitting to noise or temporary anomalies.

## 4 EXPERIMENTS

We evaluate our approach across a diverse set of data types and tasks, including age prediction, image classification, and road segmentation in the visual domain, as well as aerodynamics prediction using 3D data and tabular data experiments. In all these scenarios, we first train the model using the supervised approach of Section 3.1 and then perform the test-time training of Section 3.2. For each task, we design an OOD data subset for evaluation. The OOD data is divided into several levels, with higher levels representing data that is progressively further from the training distribution. In each experiment, we observe how quickly the model's performance degrades as the level of OOD-ness increases. We have not found test-time adaptation baselines matching the TTT problem setup for any of the tasks except image classification. So we provide the comparative numbers in this case and, for all cases, we compare against the performance of the vanilla non adaptive model. In all cases we used published, common, strong models that are SotA or close to it. Across all scenarios, our method degrades slower than the vanilla network baseline.

### 4.1 TABULAR DATA

Tabular data consists of numerical features and corresponding continuous target values for regression tasks from the UCI tabular datasets (Bay et al., 2000). They are widely used in machine learning research for benchmarking regression models. In our case, we use The Boston Housing dataset describes housing prices in the suburbs of Boston, Massachusetts. It includes various features related to socioeconomic and geographical factors that influence housing prices. We take a test set and gradually apply random feature zeroing with increasing probabilities of 5%, 10%, 15%, and 20% (4 mentioned levels of OOD). This random feature dropping simulates out-of-distribution (OOD) data by progressively altering the input features, making the data less similar to the original training distribution. As the probability of feature dropping increases, the data becomes more OOD, which lowers the model's accuracy. The trained model is a simple Multi-Layer Perceptron (MLP) optimized using the Adam optimizer, and we observe that IT$^3$ consistently degrades less compared to the vanilla baseline across all OOD levels as depicted in Fig. 2.

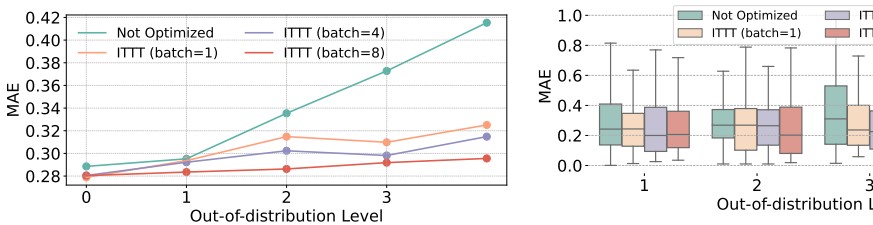

Figure 2: **UCI Results on OOD inputs**: The plots illustrate the performance of IT$^3$ compared to a vanilla model across different OOD levels. **Left**: The mean absolute error (MAE) shows that ITTT outperforms the vanilla model, retaining performance better as the data shifts further from the training distribution. **Right**: The box plot for car data shows the distribution of MAE at various OOD levels, where ITTT with different batch sizes ([batch=1, batch=4, batch=8]) degrades less compared to the Not optimized baseline. Larger batch sizes preserve performance more effectively.

### 4.2 CIFAR

We conducted similar experiments using the CIFAR-10 (Krizhevsky et al., 2014) dataset, selecting CIFAR-C (Hendrycks & Dietterich, 2019) as the out-of-distribution (OOD) data. CIFAR-C contains the same images as CIFAR-10 but with various common corruptions, such as Gaussian noise, blur, and contrast variations, simulating real-world conditions. These corruptions are applied at different severity levels, allowing us to evaluate how the model's performance degrades as the data shifts further from the original CIFAR-10 distribution. For this experiment, we used the *Deep Layer Aggregation* (DLA)(Yu et al., 2018) network, known for its strong performance in image classification and robustness to overfitting. We trained the model according to the guidelines from the original DLA paper to ensure optimal results. Fig.2 shows the evaluation error on CIFAR-C at severity level 5 for different types of corruptions, following (Sun et al., 2020). As shown, IT$^3$ outperforms the vanilla model, with higher batch sizes yielding the best results. In our basic setup, batch size of

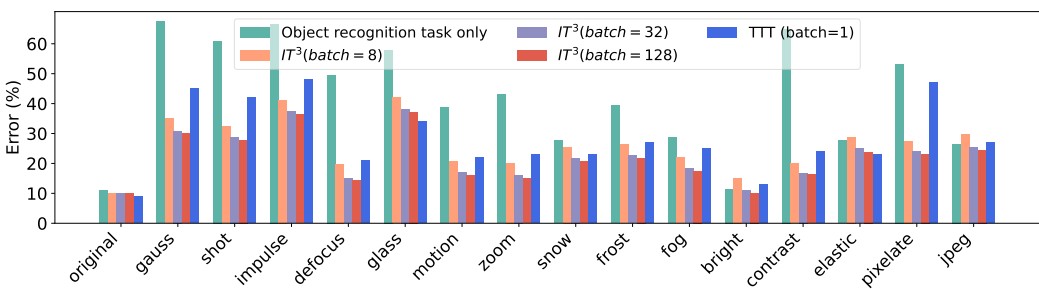

Figure 3: **Test error (%) on CIFAR-10-C with level 5 corruptions.** We compare our approaches, $IT^3$, with object recognition without self-supervision. $IT^3$ improves over the baseline and higher batch size improves even further. The comparison with TTT (Sun et al., 2020) is provided for context, but it is not a direct comparison, as TTT uses a batch size of 1. Augmentations over this single instance create a batch of 32, yet only one instance is accessed at a time.

1 does not work well. We did not explore the possibility of following Sun et al. (2020) creating a batch of augmented copies, as this would be a domain specific element, hurting the purity of our general method. For context, we add TTT results, while fully acknowledging that this is not a fair comparison as they have access to a single instance, making the batch size effectively 1, although augmented to a batch of size 32.

## 4.3 AGE PREDICTION

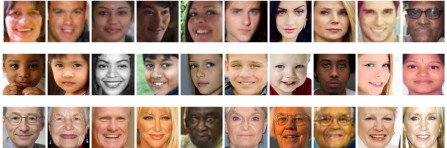

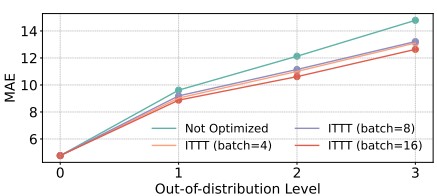

Figure 4: **Face Samples.** The **(top)** row shows training images of middle-aged individuals, while **(middle)** and **(bottom)** display images of older and younger individuals (OOD).

Figure 5: **Age mean results on OOD shapes.** The plot compares $IT^3$ to a baseline, showing better performance retention as data shifts from the training distribution.

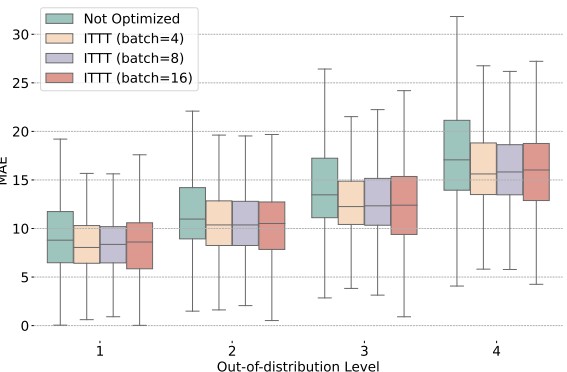

Figure 6: **Age boxplot results on OOD shapes.** Not optimized corresponds to a single model without TTT applied. $IT^3$ with [batch=4, batch=8, and batch=16] represents our method at different batch sizes. As the data shifts further from the training distribution, our method degrades less, with larger batches preserving performance more effectively.

To experiment with image-based age prediction from face images, we use the UTKFace dataset (Zhang et al., 2017), a large-scale collection containing tens of thousands of face images annotated with age information. The model is trained on face images of individuals aged between 20 and 60, while individuals younger or older than this range are considered out-of-distribution (OOD) (Fig.4). The further the age is from the 20-60 interval, the higher the OOD level we assign to it. We use a ResNet-152 backbone with five additional linear layers and ReLU activations. This architecture delivers strong accuracy, outperforming the popular ordinal regression model CORAL (Cao et al., 2020) and matching other state-of-the-art methods (Berg et al., 2021). We train our model on the UTKFace training set (limited to individuals aged 20-60) and then run inference on faces at different OOD levels. Once again, $IT^3$ significantly outperforms the non-optimized model, as shown in Figs.5 and6.

## 4.4 ROAD SEGMENTATION

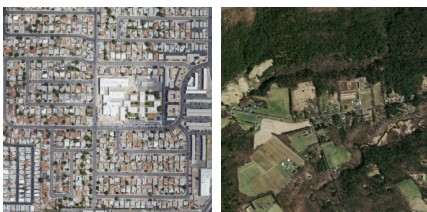

Figure 7: **Road Samples.** RoadTracer dataset **(left)** covers urban areas of six different countries while Massachusetts dataset **(right)** primarily features rural neighborhoods along with some urban areas.

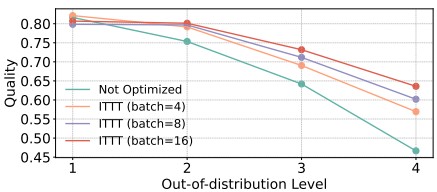

Figure 8: **Roads mean quality score on OOD images.** The plot compares $IT^3$ to the vanilla model, showing better performance retention as data shifts from the training distribution.

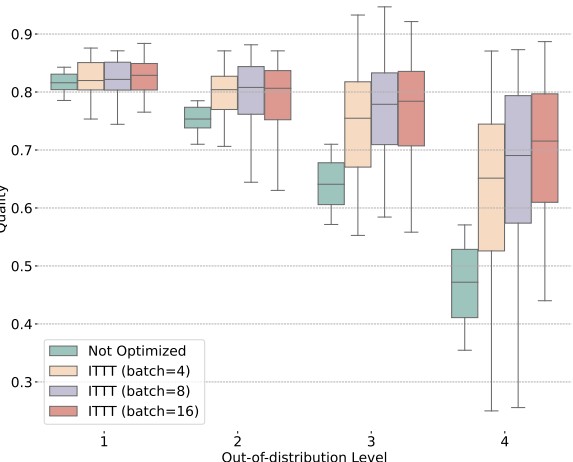

Figure 9: **Roads results on OOD images.** Not optimized corresponds to a single model without TTT applied. $IT^3$ with [batch=4, batch=8, and batch=16] represents our method at different batch sizes. As the data shifts further from the training distribution, our method degrades less, with larger batches preserving performance more effectively.

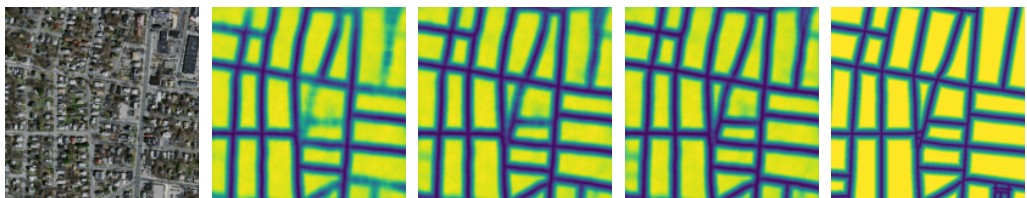

Figure 10: **Qualitative effect of $IT^3$ on Road Segmentation.** From left to right: (1) Original aerial image, (2) Not optimized output, (3) $IT^3$ output at the 5th iteration, (4) $IT^3$ output at the 15th iteration, and (5) Ground truth label. The segmentation quality improves significantly with $IT^3$ iterations, as observed in the progressively refined outputs at the 5th and 15th iterations.

Our method can be generalized to segmentation tasks as well. To demonstrate this, we consider the problem of road segmentation in aerial imagery using the RoadTracer dataset (Bastani et al., 2018). We train a DRU-Net (Wang et al., 2019), on the RoadTracer dataset.

We perform OOD experiments using Massachusetts Road dataset (Mnih, 2013) that primarily comprises rural neighborhoods, as depicted in Fig. 7. We sample 450 images, each with dimensions of 1500x1500 pixels and divide them into four groups based on the Mean Squared Error (MSE) of the segmentation outputs, effectively creating different levels of distributional shift within the sampled set. We then further train the network on these OOD subsets using the ZigZag method (Durasov et al., 2024b).

We evaluate road segmentation performance by using *Correctness*, *Completeness* and *Quality* (CCQ) metric (Wiedemann et al., 1998) which is a popular metric to evaluate delineation performance. The *Correctness*, *Completeness* and *Quality* are equivalent to precision, recall and intersection-over-union, where the definition of a true positive has been relaxed from spatial co-incidence of prediction and annotation to co-occurrence within a distance of 5 pixels. As shown in Fig. 8 and 9, $IT^3$ significantly improves the performance on OOD images.

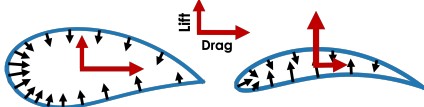

Figure 11: **Airfoil Samples.** Training and testing profiles (**left**) show reasonable aerodynamics, while OOD samples (**right**) feature rare, high lift-to-drag shapes. Black arrows indicate pressure, and red lines show lift and drag.

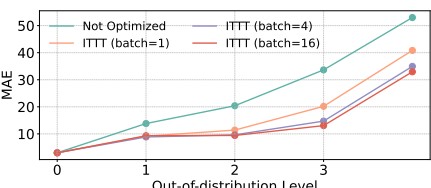

Figure 12: **Cars mean error on OOD shapes.** The plot compares ITTT to the vanilla model, showing better performance retention as data shifts from the training distribution.

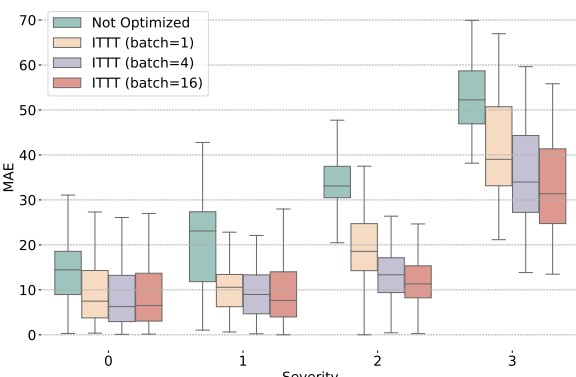

Figure 13: **Airfoil results on OOD shapes.** Not optimized corresponds to a single model without TTT applied. **ITTT** with [batch=1, batch=4, and batch=16] represents our method at different batch sizes. As the data shifts further from the training distribution, our method degrades less, with larger batches preserving performance more effectively.

### 4.5 AERODYNAMICS PREDICTION

**Wings.** Our method is versatile and can handle various types of data. To illustrate this, we generated a dataset of 2,000 wing profiles, as depicted in Fig.11, by sampling the widely used NACA parameters (Jacobs & Sherman, 1937). We used the XFoil simulator (Drela, 1989) to compute the pressure distribution along each profile and estimate its lift-to-drag coefficient, a crucial indicator of aerodynamic performance. The resulting dataset consists of wing profiles $\mathbf{x}_i$, represented by a set of 2D nodes, and the corresponding scalar lift-to-drag coefficient $\mathbf{y}_i$ for $1 \leq i \leq 2000$.

We selected the top 5% of shapes, based on their lift-to-drag ratio, as out-of-distribution (OOD) samples. The OOD levels were determined using the ground truth lift-to-drag ratio, where higher OOD levels correspond to more aerodynamically streamlined shapes. The training set includes shapes with lift-to-drag values ranging from 0 to 60, with anything beyond this threshold considered OOD and excluded from training. We then trained a Graph Neural Network (GNN) composed of 25 GMM (Monti et al., 2017) layers, featuring ELU activations (Clevert et al., 2015) and skip connections (He et al., 2016), to predict the lift-to-drag coefficient $\mathbf{y}_i$ from the profile $\mathbf{x}_i$, following the approach of (Remelli et al., 2020; Durasov et al., 2024b). As with previous experiments, $IT^3$ significantly improves performance on OOD shapes and provides more accurate predictions compared to the vanilla model, as shown in Figs. 12 and 13.

**Cars.** As for wings, we experimented with 3D car models from a subset of the ShapeNet dataset (Chang et al., 2015), which contains car meshes suitable for CFD simulations. The experimental protocol was the same as for the wing profiles, except we used OpenFOAM (Jasak et al., 2007) to estimate drag coefficients and employed a more sophisticated network to predict them from the triangulated 3D car meshes.

To predict drag associated to a triangulated 3D car, we utilize similar model to airfoil experiments but with increased capacity. Instead of twenty five GMM layers, we use thirty five and also apply skip-connections with ELU activations. Final model is being trained for 100 epochs with Adam optimizer and $10^{-3}$ learning rate.

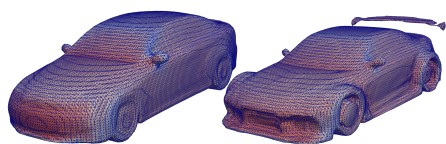

Figure 14: **Car Samples.** The car dataset comprises many regular vehicles (**left**) and a few streamlined ones (**right**), which we treat as being out-of-distribution. Red and blue denote high and low pressures respectively.

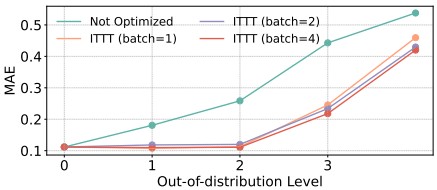

Figure 15: **Cars mean error on OOD shapes.** The plot compares ITTT to the vanilla model, showing better performance retention as data shifts from the training distribution.

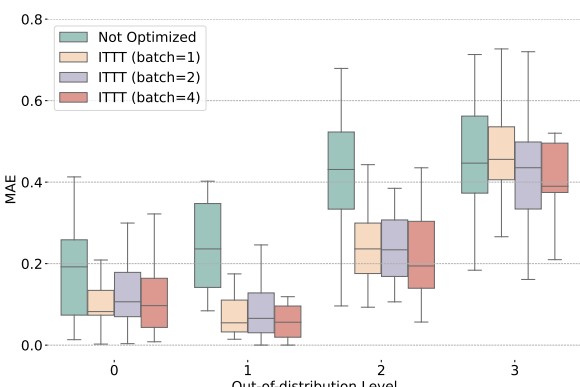

Figure 16: **Car results on OOD shapes.** Not optimized corresponds to a single model without TTT applied. **ITTT** with [batch=1, batch=2, and batch=4] represents our method at different batch sizes. As the data shifts further from the training distribution, our method degrades less, with larger batches preserving performance more effectively.

Table 1: **Qualitative result for Online IT$^3$.** We report evaluation metrics for the road segmentation task (**left**), airfoils lift-to-drag prediction (**middle**), and car drag prediction (**right**).The results suggest Online IT$^3$ enhances the performance compared to the original model. Additionally, online IT$^3$ significantly outperforms offline IT$^3$.

| METHOD | Corr | Comp | Quality | METHOD | MAE | METHOD | MAE |
|---|---|---|---|---|---|---|---|
| NOT OPTIMIZED | 55.7 | 44.3 | 39.5 | NOT OPTIMIZED | 38.2 | NOT OPTIMIZED | 0.501 |
| IT$^3$ (BATCH=4) | 55.7 | 49.1 | 46.4 | IT$^3$ (BATCH=1) | 37.6 | IT$^3$ (BATCH=1) | 0.446 |
| IT$^3$ (BATCH=8) | 58.1 | 52.0 | 48.5 | IT$^3$ (BATCH=4) | 37.5 | IT$^3$ (BATCH=2) | 0.424 |
| IT$^3$ (BATCH=16) | 57.3 | 52.7 | 48.7 | IT$^3$ (BATCH=16) | 37.4 | IT$^3$ (BATCH=4) | 0.412 |
| IT$^3$ (ONLINE) | **77.5** | **79.8** | **69.8** | IT$^3$ (ONLINE) | **34.1** | IT$^3$ (ONLINE) | **0.385** |

## 4.6 ONLINE IT$^3$

We test our proposed online variation on several tasks. Naturally, when the distribution remains constant (although shifted from the training distribution) we expect superior results w.r.t. the offline setup, as our model keeps being trained on the new distribution. A way to better test constant adaptation over time, is to have a constantly changing distribution. We test IT$^3$ on an increasing corruption/OOD level. We see in all cases that the online variation of IT$^3$ performs significantly better than the basic anchored variation.

**Road segmentation**: Building upon our previous road segmentation experiments, we further evaluate the effectiveness of online IT$^3$. In the online IT$^3$ setup, OOD samples are ranked based on their mean squared error (MSE) loss when passed through the vanilla network. We begin by selecting the samples with low MSE loss, as these are closer to the training distribution given the network's strong performance on them. Gradually, we introduce samples with progressively higher MSE loss, smoothly shifting between distributions and thereby allowing the model to adapt effectively to a range of OOD samples. As in previous experiment, we use DRU-Net trained on the RoadTracer dataset as vanilla model and 890 images are sampled from Massachusetts dataset as OOD images.

Firstly, the vanilla network is tested on the Massachusetts dataset without any additional fine-tuning. We then apply online IT$^3$ during inference to adapt the model to the OOD distribution as new data is presented. We evaluate the segmentation performance using the *Correctness*, *Completeness*, and *Quality* metrics, as described previously. Table 1 (left) summarizes the results. The application

of IT$^3$ improved the performance over the initial network and the online IT$^3$ method significantly outperforms the offline IT$^3$.

**Aerodynamics**: Similarly, we conducted online experiments for airfoils lift-to-drag prediction and for car drag prediction. We set the data stream s.t. that OOD shapes appear in an increasing aerodynamic properties, modeling a continuous domain shift in the data. As with the segmentation results, the online version significantly outperforms both the offline version and the original network, as shown in Tabs.1 (middle and right).

## 5 LIMITATIONS:

While global, IT$^3$ lacks domain expertise. Within the domains we experimented with, we are aware only of computer-vision algorithms that adhere to the restrictive problem setup. However, it is likely possible to implement domain specific methods based on self-supervision that can outperform IT$^3$.

In addition, we found that for some domains it is hard to apply IT$^3$ on single instances without also using additions that require domain expertise or access to training data. This is most common for domains where the information within a single input is limited. The reason for that is that in contrast to self-supervised auxiliary tasks, our TTT objective is based on predictions without independent information on the input data.

## 6 CONCLUSION

We have proposed an approach to test-time-training that relies on enforcing idempotence as new samples are being considered to effectively handle domain shifts. The method is generic and we have demonstrated that it is effective in a wide range of domains without requiring domain-specific knowledge, which sets it apart from other state-of-the-art methods.

In future work we plan to pursue the challenge of realistic online continual learning, where there is no pre-training at all and the data arrives in streams, sometimes with labels and sometimes not. We believe IT$^3$ can be adapted to such a setup across many different streaming modalities, which would make it extremely useful in real-world scenarios.

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

## A  APPENDIX

You may include other additional sections here.

### A.1  ADDITIONAL ROAD SEGMENTATION EXPERIMENTS

In order to further evaluate our method, we perform an additional OOD experiments using the test set of the RoadTracer dataset itself. We select cities from the RoadTracer test dataset that are not part of the RoadTracer training set and treat them as OOD samples. We divide the selected set into four groups based on the Mean Squared Error (MSE) of the segmentation outputs, effectively creating different levels of distributional shift within the test set. We then further train the network on these OOD subsets. In line with our other experiments, we demonstrate that applying $IT^3$ significantly improves segmentation performance on these OOD samples. The quantitative results of this experiment can be seen in Fig. 18 and 19.

702
703
704
705
706
707
708
709
710
711
712
713
714
715
716
717
718
719
720
721
722
723
724
725
726
727
728
729
730
731
732
733
734
735
736
737
738
739
740
741
742
743
744
745
746
747
748
749
750
751
752
753
754
755

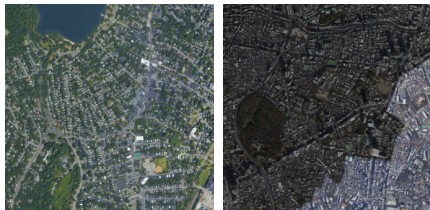

Figure 17: **Road Samples.** RoadTracer train dataset **(left)** includes urban areas of cities in US. From RoadTracer test dataset, we selected images of cities that are not included in train dataset as OOD samples **(right)**.

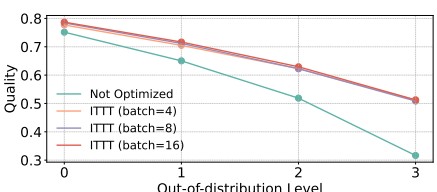

Figure 18: **Roads mean quality score on OOD images.** The plot compares IT$^3$ to the vanilla model, showing better performance retention as data shifts from the training distribution.

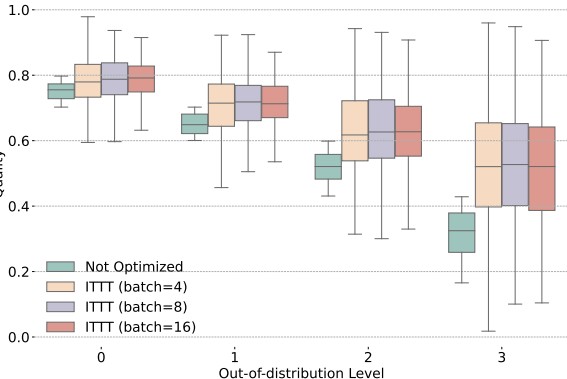

Figure 19: **Roads results on OOD images.** Not optimized corresponds to a single model without TTT applied. IT$^3$ with [batch=4, batch=8, and batch=16] represents our method at different batch sizes. As the data shifts further from the training distribution, our method degrades less, with larger batches preserving performance more effectively.

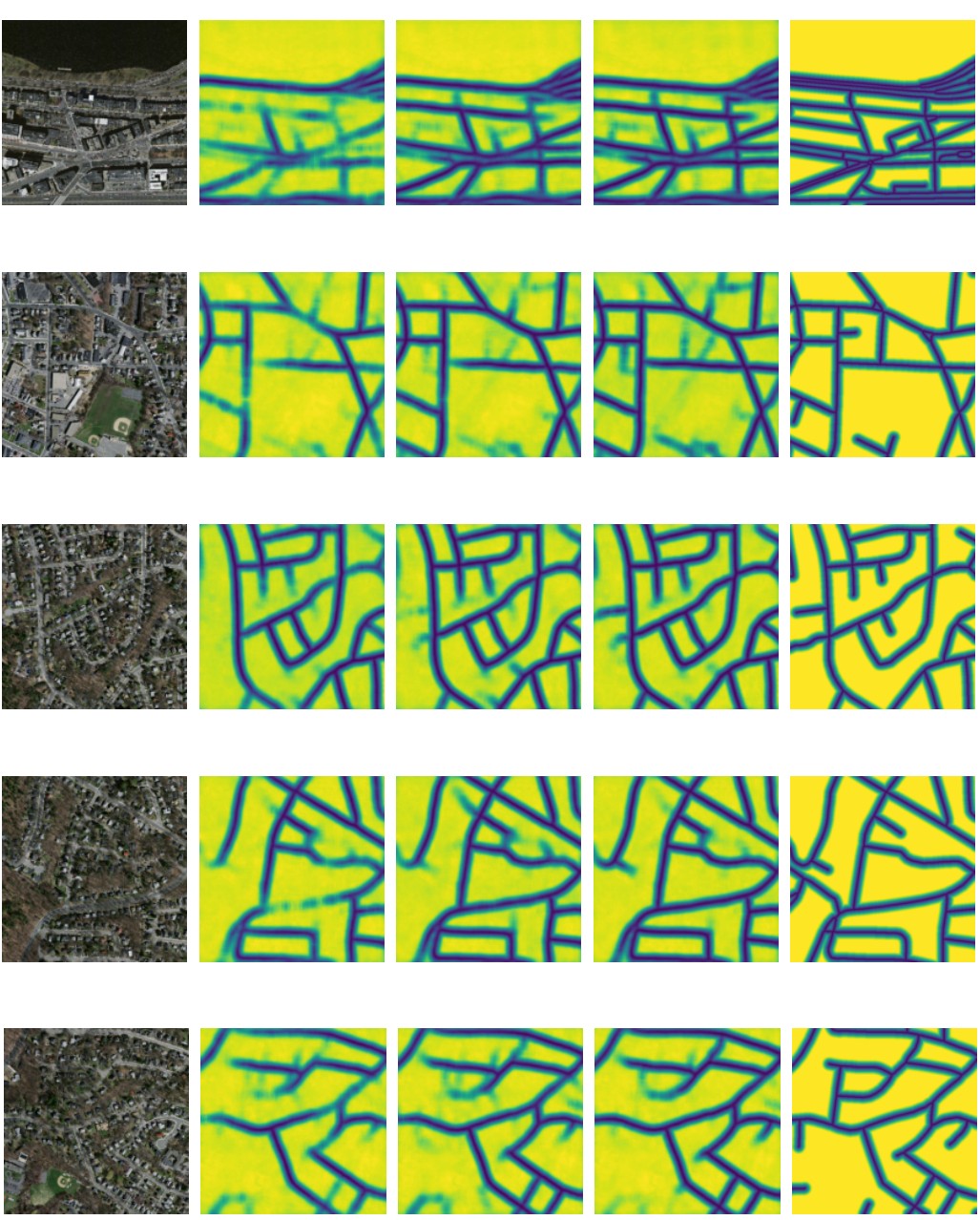

Figure 20: **Additional qualitative results of IT³ on Road Segmentation.** From left to right: (1) Original aerial image, (2) Not optimized output, (3) IT³ output at the 5th iteration, (4) IT³ output at the 15th iteration, and (5) Ground truth label. The segmentation quality improves significantly with IT³ iterations, as observed in the progressively refined outputs at the 5th and 15th iterations.

