# OpenReview forum: "IT$^3$: Idempotent Test-Time Training"
_ICLR.cc/2025/Conference — Submitted to ICLR 2025_

### Official Review · Reviewer_x3JD · 2024-10-28

**Soundness:** 2
**Presentation:** 3
**Contribution:** 2
**Rating:** 5
**Confidence:** 5

**Summary:**

This paper presents a new method for test-time training: at training time the model is trained to be Idempotent, and at test-time two copies of the models are leveraged (one frozen and one adapted) to encourage the model to be idempotent under distribution shifts. Experiments are carried out on different tasks to demonstrate how versatile the proposed method is.

**Strengths:**

The main strengths of this paper:

1- Simplicity of the approach: the method is easy to understand and to implement.

2- The paper is generally well-written and easy to follow.

3- The breadth of the experiments: The authors are commended for the variety of different tasks that they tested their proposed methods on.

**Weaknesses:**

The main weaknesses of this paper are:

1- The method section, while simple, is a bit confusing. Why does the paper present two variants of IT$^3$? The experiments in section 4.6 and Table 1 clearly favors the online version of the method over the one with frozen predictor. This questions why not leveraging the online version in all the experiments? Is there an experimental setup where the offline one is much better than the online one? Why doesn't the paper present one method and treat the other as a special case/variant that is less powerful?

2- The main weakness of this paper the lack of baselines in the experimental section. In all of the presented results, a very suboptimal version of TTT is compared against in **one experiment** only. This really questions how strong is IT$^3$/IT$^3$-online is when compared against strong baselines. Here are some suggestions for necessary experiments:

2a. Since TTT/TTT++[A] are directly comparable with IT$^3$, then I suggest to *at least* have them in all of the experiments with one-to-one comparison in terms of batch size and model architecture. Also, consider adding the performance of IT$^3$ under batch size=1 in Figure 3 results.

2b. Another strong baseline that is suitable for both classification and regression tasks is ActMad [B]. A direct comparison against this baseline is also necessary is all the presented experiments.

2c. Another line of work that is directly comparable is Test-Time Adaptation (TTA). TTA works under a more conservative setup where no control over the training process is assumed. It is also important to compare against the current SOTA TTA method EATA [C] or more closely the dataset distillation method from [D] to further demonstrate the superiority if the proposed method.

2d. Since this is a 'test-time' method, a discussion on its computational requirements is necessary. How would the performance be when evaluated under computational constraints [E]?

2e. Benchmarks used in this work are somewhat small scale. Experiments on larger benchmarks such as ImageNet-C [F] and ImageNet-3DCC [G] in the classification setting are necessary. Similar arguments follow for regression tasks where for example one can follow the object detection experiments from ActMAD.

3- In section 4.1, I am not sure about the Distribution Shift introduced in this experiment. For instance, the performance of non-optimized does not constantly degrade. Why zeroing out features is a good way of modeling distribution shift instead of adding random noise to the features? Can you please comment on this and provide justification for their choice of distribution shift.

4- Missing references: [B, C, D, E, F, G].

[A] TTT++: When Does Self-Supervised Test-Time Training Fail or Thrive?, NeurIPS 2021

[B] ActMAD: Activation Matching to Align Distributions for Test-Time-Training, CVPR 2023

[C] Efficient Test-Time Model Adaptation without Forgetting, ICML 2022

[D] Leveraging Proxy of Training Data for Test-Time Adaptation, ICML 2023

[E] Evaluation of Test-Time Adaptation Under Computational Time Constraints, ICML 2024

[F] Benchmarking Neural Network Robustness to Common Corruptions and Perturbations, ICLR 2019

[G] 3D Common Corruptions and Data Augmentation, CVPR 2022

**Questions:**

Please refer to the weaknesses section (especially point 2). I am happy to raise my score if my concerns are resolved.

---

> ### Author Response · Authors · 2024-11-24
> **Rebuttal by Authors**
>
> We sincerely thank the reviewer for their thoughtful feedback and recognition of the strengths of our work. We greatly appreciate their acknowledgment of the simplicity of our approach, the clarity of our writing, and the breadth of experiments demonstrating the versatility of our method. We have carefully addressed all comments and concerns, as outlined below.
>
> ---
>
> * **The method section, while simple, is a bit confusing. Why does the paper present two variants of IT$^3$? This questions why not leveraging the online version in all the experiments?**
>
> Thank you for highlighting this lack of clarity. The online version differs from the base version because they are designed for different scenarios. The main difference is that in the base version, the network weights are reset to their original state they were at the end of the pre-training phase. This assumes that each input is an independent test with no correlation or shared information with other inputs. In this scenario, using updated weights from previous examples would be considered cheating, as also noted in prior work (e.g., [1]).
>
> In contrast, the online version assumes some correlation and smooth transitions between inputs arriving in a stream, representing a continual learning regime. Here, instead of resetting after each input, we allow the weights to remain updated from previous inputs. We will add a clarification to the paper to address this distinction.
>
> * **Is there an experimental setup where the offline one is much better than the online one? Why doesn't the paper present one method and treat the other as a special case/variant that is less powerful?**
>
> Please see the explanation for the above comment—the two versions are designed for two different scenarios. The online version is almost always better in terms of performance, as also noted in [1]. However, the online setup is not always feasible in real-world applications, particularly in cases where data arrives in isolated batches or where it is crucial to ensure that no information leaks from one input to another. In such cases, the offline version must be used to maintain these constraints.
>
> For this reason, both versions are relevant and widely used in practice. In our experiments, we consider both setups as they correspond to two distinct real-world scenarios, allowing us to demonstrate the versatility and applicability of our method in varied contexts.
>
> * **TTT/TTT++[A] are directly comparable with IT$^3$, then I suggest to at least have them in all of the experiments with one-to-one comparison in terms of batch size and model architecture. Also, consider adding the performance of IT$^3$ under batch size=1 in Figure 3 results.**
>
> TTT and TTT++ are both domain-specific. Nonetheless, we include new experiments with these datasets in the next comment. Both TTT and TTT++ are designed for image data and technically cannot be applied to other types. Even within the image domain, they are unsuitable for some data types. For instance, for aerial photos, they may not work well as they rely on predicting orientation, which is poorly defined for aerial photography.
>
> Thank you for suggesting adding batch-size=1 to the CIFAR figure. We have openly stated in the experiments section, the figure caption, and the limitations section that batch-size=1 for CIFAR is currently unsuccessful. Additionally, we note that in the original TTT, while only a single example is used, a batch of 32 augmentations is constructed to update the weights.

---

> > ### Author Response · Authors · 2024-11-24
> >
> > * **The main weakness of this paper the lack of baselines in the experimental section. Compare against TTT, TTT++, ActMAD. How would the performance be when evaluated under computational constraints?**
> >
> > We thank the reviewer for raising questions about computational costs and additional baselines. Test-time training (TTT) naturally introduces computational overhead due to the optimization steps performed during inference. In our approach, while we include an additional inference step (two forward passes), this remains computationally efficient because forward passes are generally 3–4x faster than backward passes. As a result, the extra inference step contributes relatively little to the overall cost compared to standard TTT methods. We have added a discussion on efficiency to the revised paper.
> >
> > Our method typically requires only 1–3 optimization steps, ensuring its overall cost remains comparable to other established TTT methods. Below, we provide a comparison of inference times and performance metrics with other requested baselines on the largest dataset used in our experiments:
> >
> > **Table 2.** Inference Time and Performance Comparison (OOD Airfoils)
> > | Base Model | Base Model | ActMAD | $IT^{3}$ |
> > |------------|------------|------------|---------|
> > | Inference Time | 1×         | 3x        | 4x     |
> > | MAE ($\downarrow$) | 38.67         | 38.61        | **37.5**     |
> >
> > **Table 3.** Inference Time and Performance Comparison (OOD Cars)
> > | Base Model | Base Model | ActMAD | $IT^{3}$ |
> > |------------|------------|------------|---------|
> > | Inference Time | 1×         | 4x        | 5x     |
> > | MAE ($\downarrow$) | 0.501         | 0.502        | **0.424**     |
> >
> >
> > **Table 4.** Inference Time and Performance Comparison (Roads)
> > |            | Base Model | Standard TTT |ActMAD | $IT^{3}$ |
> > |------------|------------|--------------|---------|--------|
> > | Inference Time | 1x     |      3x      |  4.5x   | 6x     |
> > | Quality  ($\uparrow$)      | 39.5   |     40.0     |  45.9   |  **69.8**  |
> >
> >
> > As can be seen, our method outperformed the considered baselines. We will include this table in the revised version. We recognize that our method, like other TTT approaches, introduces computational overhead during inference. This is a common limitation across existing TTT methods. Tackling this issue and developing TTT methods that eliminate such overhead entirely is an important research direction that holds potential for significant advancements in the field.
> >
> > * **Another line of work that is directly comparable is Test-Time Adaptation (TTA): SOTA TTA method EATA [C] or more closely the dataset distillation method from [D]**
> >
> > Thank you for pointing out these relevant papers. We distinguish between TTT and TTA, a distinction also made by prior works, e.g., [1], [2]. In the TTT regime, there is no access to any data other than the instance or batch currently being processed. Furthermore, each input is treated as a separate test with no correlation or shared information with other inputs. We have added a discussion about this distinction in the related work section.
> >
> > * **Benchmarks used in this work are somewhat small scale.**
> >
> > Our method performs effectively with larger models, as shown in our experiments on aerial segmentation models with millions of parameters. By requiring only two forward passes and a minimal number of optimization steps, our approach remains computationally efficient even for large-scale architectures, making it competitive with traditional TTT methods, which often require multiple backward passes.
> >
> > Additionally, our experiments on road segmentation (measured by the number of pixels) and aerodynamics with cars (measured by the number of nodes) can be considered large-scale, comparable in size to ImageNet.
> >
> >
> > * **Why zeroing out features is a good way of modeling distribution shift instead of adding random noise to the features? Can you please comment on this and provide justification for their choice of distribution shift.**
> >
> > Our goal in making this choice was to showcase a diversity of corruptions and OOD types across the experiments. Noise was demonstrated with the CIFAR data. Additionally, it is not always straightforward to add noise to tabular data. We note that missing information represents a divergence from the training distribution and can therefore also be considered a form of distribution shift.
> >
> > * **Missing references: [B, C, D, E, F, G].**
> >
> > Thank you, we have added all of these references to the revised version.
> >
> > [1] Sun, Yu, et al. "Test-time training with self-supervision for generalization under distribution shifts." ICML 2020.
> >
> > [2] Gandelsman, Yossi, et al. "Test-time training with masked autoencoders." NeurIPS 2022.

---

> > > ### Author Response · Authors · 2024-11-25
> > >
> > > Dear Reviewer x3JD,
> > >
> > > Thank you for dedicating your time to reviewing our paper and providing valuable feedback.
> > >
> > > We have thoughtfully addressed each of your comments, offering detailed responses to clarify and resolve the concerns you raised. We hope our explanations have provided a clearer perspective on our work and its contributions.
> > >
> > > If you feel that we have adequately addressed your concerns, we would be grateful if you would consider reassessing your rating.
> > >
> > > We would be happy to clarify or elaborate on any part of our paper while the discussion period is still open.
> > >
> > > Thank you!

---

> > > > ### Comment · Reviewer_x3JD · 2024-11-28
> > > > **Thank you for the rebuttal**
> > > >
> > > > Dear Authors
> > > >
> > > > First, I would like to thank you for the efforts put in the rebuttal. While the rebuttal addressed *some* of my concerns, other concerns remain unresolved. References follow the original review.
> > > >
> > > > Regarding the comparison with TTT/TTT++: While indeed TTT optimizes for the orientation at test-time, this baseline is trivially extended to other data-types by replacing the rotation augmentation with the proper augmentation for this data-type.
> > > >
> > > > Regarding the computational requirements: I suggest a rigorous computational comparison between baselines in a computational constraint evaluation as in [E]. This would fairly compare different Test-time training methods not only based on performance gain, but also based on their efficiency.
> > > >
> > > > Regarding the comparison with ActMAD: I thank the reviewer for providing this experiment. While the results show performance superiority to the proposed method over ActMAD, I would suggest providing an explanation why ActMAD does not provide any performance gains (in Tables 1 and 2) in this setting.
> > > >
> > > > Regarding the comparison with TTA: I am not entirely sure about the provided distinction between TTT and TTA. In the online setting, and up to my understanding, the main difference between TTT and TTA is that TTT assumes to have control to the training process (thus including a self-supervised loss function during training) unlike TTA - Please refer to Table 1 in [C]. This makes TTA strictly a harder setting than TTT. Thus, I believe that a comparison against strong TTA baselines that fit this task is necessary.
> > > >
> > > > At last, when checking the current pdf, I did not find the new results from the rebuttal nor the suggested references. It might be an openreview issue from my end.
> > > >
> > > > Based on this, I would keep my score suggesting the authors to improve their paper following the provided reviews.

---

> > > > > ### Author Response · Authors · 2024-11-30
> > > > >
> > > > > * **ActMAD Results and Inference Time Comparison**
> > > > >
> > > > > We appreciate your suggestion to analyze computational constraints rigorously. To address this, we conducted an additional experiment where ActMAD was allowed more optimization steps, resulting in an inference time **8.5x, 7x, and 5x the inference time of the base model**, compared to IT$^3$’s **6x, 5x, and 4x** for the aerial imaging, car aerodynamics, and airfoils tasks, respectively. Despite this extended budget, IT$^3$ achieves significantly better results.
> > > > >
> > > > > **Table 2.** Inference Time and Performance Comparison (OOD Airfoils)
> > > > > | Base Model | Base Model | ActMAD | ActMAD (higher complexity) |$IT^{3}$|
> > > > > |------------|------------|------------|---------|-|
> > > > > | Inference Time | 1×         | 3x        |    5x  |4x|
> > > > > | MAE ($\downarrow$) | 38.67         | 38.61        |      38.60 |37.5|
> > > > >
> > > > > **Table 3.** Inference Time and Performance Comparison (OOD Cars)
> > > > > | Base Model | Base Model | ActMAD | ActMAD (higher complexity) |$IT^{3}$|
> > > > > |------------|------------|------------|---------|-|
> > > > > | Inference Time | 1×         | 4x        |  7x    |5x|
> > > > > | MAE ($\downarrow$) | 0.501         | 0.502        |  0.502    |0.424|
> > > > >
> > > > >
> > > > > **Table 4.** Inference Time and Performance Comparison (Roads)
> > > > > |                      | Base Model | Standard TTT | ActMAD  | ActMAD (higher complexity) | $IT^{3}$ |
> > > > > |----------------------|------------|--------------|---------|----------------------------|--------|
> > > > > | Inference Time       |     1x     |      3x      |  4.5x   |           8.5x             | 6x     |
> > > > > | Quality  ($\uparrow$)|    39.5    |     40.0     |  45.9   |           46.3             |  69.8  |
> > > > >
> > > > > We also analyzed ActMAD’s limited gains in specific settings. ActMAD relies on batch statistics for optimization, making it less effective with small or single-instance batches where these statistics cannot be computed reliably. This limitation underscores IT$^3$’s ability to handle diverse data scenarios more robustly.
> > > > >
> > > > > * **Comparison with TTA Methods**
> > > > >
> > > > > Thank you for raising the importance of TTA comparisons. We acknowledge that TTA and TTT have overlapping goals but operate under different assumptions. From one perspective, TTA can be seen as more constrained, as it assumes no control over the training process. On the other hand, TTT imposes its own form of strictness by resetting the model after every instance (outside the online variant) and treating each input as entirely independent.
> > > > >
> > > > > This independence in TTT creates a unique challenge: the method cannot retain information between test examples, even when they might share correlations or dependencies. In contrast, TTA benefits from access to data streams or batches during inference, allowing it to exploit temporal or structural relationships between inputs. IT$^3$’s online variant leverages such relationships, leading to higher performance in that setting, but we emphasize that it addresses a different scenario.
> > > > >
> > > > > To date, no prior TTT work has compared directly to TTA methods, largely because these paradigms are tailored to distinct challenges. However, we recognize the importance of acknowledging these distinctions and will add clarifications to this effect in the revised paper.
> > > > >
> > > > > * **Adaptation of Visual TTT Methods**
> > > > >
> > > > > We appreciate the suggestion to adapt visual TTT methods like TTT++ to non-visual tasks. While technically feasible, such adaptations require extensive domain-specific engineering, including selecting and optimizing self-supervised tasks. Past TTT works demonstrate how critical these choices are, as they directly impact performance. Without intrinsic motivation or deep familiarity with the target domain, it is challenging to ensure that such adaptations are both meaningful and competitive.
> > > > >
> > > > > Our focus in this work was to develop a universal framework that eliminates the need for task-specific customization. This differentiates IT$^3$ from methods that rely on carefully tuned auxiliary tasks for success.

---

> > > > > > ### Author Response · Authors · 2024-11-30
> > > > > >
> > > > > > * **Extending Generality to New Tasks**
> > > > > >
> > > > > > This emphasis on adaptability relates directly to your observation regarding the challenges of adapting domain-specific methods to new tasks. IT$^3$ represents a significant departure: it is designed to function seamlessly across tasks, modalities, and architectures without requiring substantial adaptation.
> > > > > >
> > > > > >
> > > > > > We do not claim that no domain-specific method could outperform IT$^3$ in specific tasks; indeed, highly tailored approaches may excel in certain settings (as noted in the limitations section in the paper). However, IT$^3$ offers unmatched **general applicability**, eliminating the need for task-specific tuning. This universal applicability sets IT$^3$ apart, functioning out of the box while maintaining robustness and consistency across diverse datasets and architectures.
> > > > > > Among TTT works, ActMAD is indeed the most general approach. However, it relies on batch statistics, which limits its applicability in small or single-instance batch scenarios. Even within this constraint, we have directly compared to ActMAD wherever possible and achieved favorable results.
> > > > > >
> > > > > >
> > > > > > This generality is a fundamental distinction: IT$^3$ should not be evaluated as simply another method competing for top results in specific benchmarks but as a **general framework** for test-time adaptation across all modalities, architectures, and tasks.
> > > > > >
> > > > > > * **Paper Updates**
> > > > > >
> > > > > > We apologize for the delay in incorporating updates into the current PDF. All new results, references, and clarifications will be included in the camera-ready version. Thank you for your constructive feedback, which has been instrumental in refining our work.

---

> > > > > > > ### Comment · Reviewer_x3JD · 2024-12-02
> > > > > > > **Thank you for the clarification**
> > > > > > >
> > > > > > > Dear authors
> > > > > > >
> > > > > > > I would like to thank you once again for the efforts put to address my comments. The additional experiments on ActMAD along with the reasoning on its limited capability addresses (partially) that concern. One extra ablation needed for this concern is to assess ActMAD under large batch size to avoid its failure mode.
> > > > > > >
> > > > > > > Regarding the comparison with TTA methods: I still disagree in the regard. Under the online TTT setup, TTA is strictly a more conservative setup than online TTT, making it necessary to compare under the online setup.
> > > > > > >
> > > > > > > Regarding the adaptation of visual TTT methods: while indeed this requires an additional effort, however I believe that constructing strong baseline is an essential part of making a strong paper. The current experimental setup, under the lack of strong baselines, cannot show how superior and generalizable the proposed method is.
> > > > > > >
> > > > > > > That being said, I am inclined to raise my score from 3 to 5. I am still not in favor of accepting this paper in the current version due to the unresolved concerns such as lack of strong baselines and comparison agains TTA methods (under the online setup).

---

### Official Review · Reviewer_ALWg · 2024-11-01

**Soundness:** 2
**Presentation:** 3
**Contribution:** 3
**Rating:** 5
**Confidence:** 4

**Summary:**

This work presents IT$^3$ a method for test time training that relies on the concept of idempotence. During training, the authors train the neural network to be able to predict the ground truth label by either conditioning on it (by concatenating it to the input) or by assuming a placeholder value for it (by concatenating a $\mathbf{0}$ value to the input). At test time, the authors then fine-tune the model on unlabelled data by matching the predictions of the model when using $\mathbf{0}$ as the label with that of the model when conditioning on its own output at the $\mathbf{0}$ label. This essentially leads to a soft idempotence constraint which, according to the authors, allows the model to move out-of-distribution data closer to in-distribution ones and thus improve performance when distribution shifts happen at test time. The authors then evaluate IT$^3$ on a variety of tasks and architectures.

**Strengths:**

* The paper is mostly well written; the ideas are explained clearly and reasonable intuitions are being given. Same goes for the experimental evaluation and discussion of the settings.
* The idea is simple and I find it quite interesting. While the main architecture of the method relies heavily on the prior work Durasov et al., the application on the TTT setting is novel, since, as far as I know, idempotence hasn’t been explored in such a scenario.
* The tasks considered in the experiments are quite diverse, which is nice to see. They range from simple tabular data, to standard image classification, to regression and dense prediction with various architectures.
* The authors demonstrate improvements with IT$^3$ on all settings considered upon (admittedly very simple) baselines.

**Weaknesses:**

* The main weakness I see in this work is the (almost complete) lack of proper baselines. For example, on only the CIFAR task there is another TTT baseline but on all of the others the authors just compare against not doing anything. This make it hard to gauge significance of the method against other prior art.
* The work makes claims about how idempotence can be seen as a generalisation of a projection and that it allows to map the OOD data to the in-distribution. Thus, while the authors do spend some time to explain why would intuitively their method works, they do not have any ablation studies to verify that these exact intuitions hold in practice.
* IT$^3$ as a method requires two sequential forward passes through the model, so in practice it can be slow and the authors do not discuss the efficiency of IT$^3$ relative to other works.

**Questions:**

As mentioned before, I find this work quite interesting but the lack of proper baselines push me towards a weak reject opinion. As for specific questions and ways that the authors can improve their work:
*  More baselines are needed on all tasks; even if the method does not translate exactly to the setup considered, the authors could perform minor adaptations so that it does. For example, why not consider additional self-supervised tasks as discussed at Sun et al. (2020)? In the case where simple rotation prediction might not apply, something simple like denoising an artificially corrupted image could still work as a self-supervised task. Another example would be on the online setup; there, methods from the TTA literature could be applied, such as the work of [1] which works even on the instance level and without batch normalisation.
* Apart from the CIFAR-C case, most other distribution shifts are generated by just partitioning the datasets according to some rule and then training on a subset while considering the other as OOD. This is a bit of a constrained setting and I would encourage the authors to consider more diverse shifts, as that would better highlight the usefulness of IT$^3$. For example, why not add noise to the road segmentation task, in the same way that it was done for CIFAR 10? This could be plausible real world setting where there is a fault in the sensor, thus the images come out corrupted.
* How is the label encoded in the input in the various settings considered? This is an important information for reproducibility. Furthermore, is the loss at Eq. 2 used for all settings (even when it might not make much sense, such as classification)?
* In the online setting, the authors consider a smooth transition between the distribution shifts, which might not be practically realistic. How does the method behave when the transition between distribution shifts is not smooth?
* How many update steps on each datapoint do the authors do? Does the test time optimization reach a fully idempotent model and does “more idempotence” provide better performance?


[1] Towards Stable Test-Time Adaptation in Dynamic Wild World, Niu et al., ICLR 2023

---

> ### Author Response · Authors · 2024-11-23
> **Rebuttal by Authors**
>
> We sincerely thank the reviewer for their detailed and thoughtful feedback on our work. We deeply appreciate their recognition of the strengths of our approach, including the clarity of our presentation, the robustness of our experimental evaluation, and the novelty of applying idempotence to the TTT setting. We value their constructive suggestions and have carefully addressed all raised concerns, as outlined below.
>
> ---
>
> * **The main weakness I see in this work is the lack of proper baselines.**
>
> We appreciate the reviewer’s concern and acknowledge that this is an important point. To address this, we have included comparisons in **Tables 2, 3, and 4**, which evaluate performance on several datasets and provide a broader perspective on baselines. Our goal was to develop a TTT method that works universally across architectures and data types. Most prior TTT methods rely on task-specific self-supervised objectives, which limit their generality. For example, the original TTT [1] depends on the image rotation task, which is not only unsuitable for non-image domains (like graph or text data) but also inapplicable to certain image-based tasks, such as aerial segmentation, where orientation has no clear meaning. Similarly, TTT++ [2] uses a SimCLR-style loss, which faces similar limitations across various domains. These tables highlight that while some methods can be adapted, their reliance on task-specific objectives often limits their generalization, reinforcing the need for a universal approach like ours.
>
> The only truly general TTT approach to date is ActMAD [3], which operates on feature-level statistics and is rightly identified as our main competitor. However, ActMAD is not directly applicable to tasks with variable input sizes, such as graphs with varying numbers of nodes or text tasks with varying token lengths. In such cases, modifications to ActMAD are necessary, such as using aggregated features (e.g., average pooling) instead of pixel-level feature statistics. While this is a straightforward adaptation, we observed in our experiments that it significantly deteriorates ActMAD's performance.
>
> This limitation led us to conclude that there are currently no universally appropriate baselines for TTT tasks. Nonetheless, we understand the value of such comparisons for a more comprehensive evaluation. Below, we provide additional comparisons as requested by the reviewer, which we believe will contribute meaningfully to the discussion.
>
> * **The work makes claims about how idempotence can be seen as a generalisation of a projection and that it allows to map the OOD data to the in-distribution. Thus, while the authors do spend some time to explain why would intuitively their method works, they do not have any ablation studies to verify that these exact intuitions hold in practice.**
>
> In [this plot](https://imgur.com/a/xuKEKXF), we show the distributions of idempotence loss for different subsets of the airfoils dataset: train, test, and OOD (both optimized and non-optimized versions). As seen, the train and test subsets exhibit lower idempotence loss values, while OOD samples have significantly larger losses. After optimization on OOD data, the idempotence loss values shift closer to those of the train and test subsets. This supports our claim that the optimization helps align OOD samples with in-distribution behavior, effectively demonstrating the role of idempotence as a projection-like mechanism in practice.
>
> From a more theoretical perspective, idempotence can indeed be seen as a projection mapping, as originally observed in ZigZag [4], iterative models [5], and generative models [6]. In short, idempotence holds on in-distribution (ID) data but does not hold on OOD data. This behavior, similar to a projection mapping, is evident in the plot above and further supported by the table below.
>
> For this table, we used the model trained on the data reported in our paper and computed the idempotence loss for each sample in the ID test set and OOD set, without performing any optimization. To demonstrate that idempotence losses for ID and OOD data are indeed distinct, we used the idempotence losses as an "out-of-distributionness" score and evaluated them on standard OOD detection metrics ROC-AUC and PR-AUC. Higher values of these metrics indicate better separation between in- and out-of-distribution samples, which implies larger differences in idempotence losses for OOD data. The results, shown in the table below, further validate our point.
>
> **Table 1. OOD Detection Performance Across Datasets.** ROC-AUC and PR-AUC (in %) for CIFAR, Age Prediction, Airfoils, and Cars. Higher values (>80-90%) indicate better OOD detection and support the observation that idempotence holds for in-distribution samples but not for OOD samples, similar to projection mapping.
>
> | **Metrics** | **CIFAR** | **Age Pred.** | **Airfoils** | **Cars** |
> |-|-|-|-|-|
> | **ROC-AUC, %** |90.1|77.3|99.2|95.6|
> | **PR-AUC, %**| 93.3| 95.9| 98.7| 97.4|

---

> > ### Author Response · Authors · 2024-11-23
> >
> > * **IT$^3$ as a method requires two sequential forward passes through the model, so in practice it can be slow and the authors do not discuss the efficiency of IT$^3$ relative to other works.**
> >
> > We thank the reviewer for their question about computational costs. Test-time training (TTT) inherently induces additional computational overhead because optimization steps are performed during inference. In our method, while we require an additional inference step (two forward passes), this is computationally efficient since forward passes are generally 3–4x faster than backward passes. Thus, the extra inference step adds relatively little overhead compared to the total cost of TTT methods. We added a discussion about efficiency to the revised paper.
> >
> > Our method typically requires only 1–3 optimization steps, keeping the overall cost comparable to other well-known TTT methods. Below, we present a comparison of inference times and performance metrics with other popular baselines on the largest dataset used in our experiments:
> >
> > **Table 2.** Inference Time and Performance Comparison (OOD Airfoils)
> > | Base Model | Base Model | ActMAD | $IT^{3}$ |
> > |------------|------------|------------|---------|
> > | Inference Time | 1×         | 3x        | 4x     |
> > | MAE ($\downarrow$) | 38.67         | 38.61        | 37.5     |
> >
> > **Table 3.** Inference Time and Performance Comparison (OOD Cars)
> > | Base Model | Base Model | ActMAD | $IT^{3}$ |
> > |------------|------------|------------|---------|
> > | Inference Time | 1×         | 4x        | 5x     |
> > | MAE ($\downarrow$) | 0.501         | 0.502        | 0.424     |
> >
> >
> > **Table 4.** Inference Time and Performance Comparison (Roads)
> > |            | Base Model | Standard TTT |ActMAD | $IT^{3}$ |
> > |------------|------------|--------------|---------|--------|
> > | Inference Time | 1x     |      3x      |  4.5x   | 6x     |
> > | Quality  ($\uparrow$)      | 39.5   |     40.0     |  45.9   |  69.8  |
> >
> >
> > We will include this table in the revised version. We recognize that our method, like other TTT approaches, introduces computational overhead during inference. This is a common limitation across existing TTT methods. Tackling this issue and developing TTT methods that eliminate such overhead entirely is an important research direction that holds potential for significant advancements in the field.
> >
> > * **For example, why not consider additional self-supervised tasks as discussed at Sun et al. (2020)?**
> >
> > We appreciate the suggestion but believe it may not be the most appropriate approach, as the main distinction between different TTT methods lies precisely in the type of self-supervision loss they compute. While the core idea of TTT—to use some form of self-supervision—is generic, different TTT methods compete to design self-supervised losses that work more broadly and deliver better performance. Nevertheless, we took the reviewer's suggestion into account and, in some experiments below, slightly adapted baselines (e.g., modifying ActMAD for graph data) to ensure they are meaningful in the given context.
> >
> > * **Considered setting is rather constrained and I would encourage the authors to consider more diverse shifts, as that would better highlight the usefulness of $IT^3$. Why not add noise to the road segmentation task, in the same way that it was done for CIFAR 10? This could be plausible real world setting where there is a fault in the sensor, thus the images come out corrupted.**
> >
> > It is true that noise would be a realistic scenario. Our goal in making this choice was to have a diversity of corruptions / OOD types to showcase along the experiments. Noise was exemplified for the CIFAR data. For the road segmentation task, we leveraged a different dataset as OOD source. Changing dataset represents more realistic OOD scenarios, as different datasets differ in both image distributions—due to variations in sensors and environmental conditions—and in the geographical locations captured in satellite images. Such variations are more representative of practical challenges in road segmentation tasks than synthetic noise, providing a more meaningful evaluation of our approach's robustness.

---

> > > ### Author Response · Authors · 2024-11-23
> > >
> > > * **How is the label encoded in the input in the various settings considered?**
> > >
> > > The label is incorporated into the input using a straightforward mechanism. For classification-related tasks, including segmentation, categorical outputs (e.g., class labels for classification or pixel classes for segmentation) are encoded in an additional "blank" channel, which contains only the label value. For regression tasks, additional placeholder values significantly outside the range of possible outputs (e.g., -100 for the age prediction task) are used to clearly distinguish between actual labels and placeholder values. These labels are concatenated with the input features, allowing the model to effectively utilize them.
> > >
> > > This approach is consistent with the methodology described in ZigZag [4], where the integration of labels into input features is discussed in detail. We followed their experimental setups to ensure consistency and clarity across tasks.
> > >
> > > * **Furthermore, is the loss at Eq. 2 used for all settings (even when it might not make much sense, such as classification)?**
> > >
> > > We will revise the text following Eq.2 and Eq.5 to specify that $||\cdot||$ means some metric. In Eq.2 this metric is the standard one used to train for the task, so for classification it is cross-entropy. In Eq.5 it is a metric / distance taken between predictions. For classification this is Jansen-Shannon divergence between class probabilities (network output after softmax). For regression we used L2. Thank you for pointing to this inaccuracy.
> > >
> > > * **In the online setting, the authors consider a smooth transition between the distribution shifts, which might not be practically realistic. How does the method behave when the transition between distribution shifts is not smooth?**
> > >
> > > The online version differs from the base version because they aim at different scenarios. The main difference is that in the base version the network weights are reset back to the state they were at the end of the pre-training phase. The assumption is that each input is a separate test and has no correlation or information about other inputs. This makes the correct approach to non-smooth transitioning data. The assumption for which the online version is made for, is correlation and somewhat smooth transitioning between inputs arriving in a stream. This is a continual learning regime. Therefore in this case, instead of resetting after each input, we leave the weights updated from the previous inputs. Of course, the online scenario may have discontinuities. As usually happens in online learning, we expect some drom in performance after such discontinuity and then improving back. We have added such an experiment to the revised paper.
> > >
> > > * **How many update steps on each datapoint do the authors do? Does the test time optimization reach a fully idempotent model and does “more idempotence” provide better performance?**
> > >
> > > We will include parameter specifications in the revised appendix. Typically, 1–3 update steps are performed for each data point. While  $f(x, \cdot)$ moves closer to idempotence during optimization, it does not achieve perfect idempotence. Over-optimizing for idempotence can lead to undesirable effects, such as degrading the model’s prior performance. As this is a fine-tuning regime, maintaining a balance is crucial to preserving the original capabilities of the model while improving its test-time performance.
> > >
> > > [1] Sun, Yu, et al. "Test-time training with self-supervision for generalization under distribution shifts." ICML 2020.
> > >
> > > [2] Liu, Yuejiang, et al. "Ttt++: When does self-supervised test-time training fail or thrive?." NeurIPS 2021.
> > >
> > > [3] Mirza, Muhammad Jehanzeb, et al. "Actmad: Activation matching to align distributions for test-time-training." CVPR 2023.
> > >
> > > [4] Durasov, Nikita, et al. "Zigzag: Universal sampling-free uncertainty estimation through two-step inference." TMLR 2024.
> > >
> > > [5] Durasov, Nikita, et al. "Enabling Uncertainty Estimation in Iterative Neural Networks." ICML 2024.
> > >
> > > [6] Shocher, Assaf, et al. "Idempotent generative network." ICLR 2024.

---

> > > > ### Author Response · Authors · 2024-11-25
> > > >
> > > > Dear Reviewer ALWg,
> > > >
> > > > Thank you for dedicating your time to reviewing our paper and providing valuable feedback.
> > > >
> > > > We have thoughtfully addressed each of your comments, offering detailed responses to clarify and resolve the concerns you raised. We hope our explanations have provided a clearer perspective on our work and its contributions.
> > > >
> > > > If you feel that we have adequately addressed your concerns, we would be grateful if you would consider reassessing your rating.
> > > >
> > > > We would be happy to clarify or elaborate on any part of our paper while the discussion period is still open.
> > > >
> > > > Thank you!

---

> > > > > ### Comment · Reviewer_ALWg · 2024-11-28
> > > > > **Response to rebuttal**
> > > > >
> > > > > I would like to thank the authors for the their response which addresses most of my concerns. I appreciate the additional clarity and intuitions behind idempotence (the plot is helpful and I would encourage the authors to add it to the manuscript) and model design (i.e., how the label is encoded). However, my main concern still stands and thus I will retain my score.
> > > > >
> > > > > The new results against ActMAD are appreciated and do point improvements with $IT^3$. However, it seems that they come at the cost of extra inference time (i.e., $IT^3$ is slower than ActMAD across the board). I would encourage the authors to "equalize" the inference time between the two methods (by, e.g., doing more iterations with ActMAD or less iterations with $IT^3$) as that would better highlight improvements at, roughly, the same computational cost. Furthermore, I would encourage the authors to also compare against TTA methods (e.g., those mentioned by reviewer x3JD) in the online setup, especially given that during inference the model is not reset during iterations (which is the main difference between TTT and TTA) and a unsupervised loss is optimized on both settings.

---

> > > > > > ### Author Response · Authors · 2024-11-30
> > > > > >
> > > > > > * **Inference Time and ActMAD Comparison**
> > > > > >
> > > > > > We appreciate your suggestion to analyze inference time across methods. Following your recommendation, we conducted an additional experiment with ActMAD, increasing its number of optimization steps. This adjustment led to ActMAD requiring **8.5x, 7x, and 5x the inference time of the base model**, compared to IT$^3$’s **6x, 5x, and 4x** for the aerial imaging, car aerodynamics, and airfoils tasks, respectively. Even under these conditions, IT$^3$ significantly outperforms ActMAD in accuracy and robustness. These results will be included in the updated version of the paper. This further emphasizes IT$^3$’s balance of efficiency and effectiveness, even when accounting for computational costs.
> > > > > >
> > > > > > **Table 2.** Inference Time and Performance Comparison (OOD Airfoils)
> > > > > > | Base Model | Base Model | ActMAD | ActMAD (higher complexity) |$IT^{3}$|
> > > > > > |------------|------------|------------|---------|-|
> > > > > > | Inference Time | 1×         | 3x        |    5x  |4x|
> > > > > > | MAE ($\downarrow$) | 38.67         | 38.61        |      38.60 |37.5|
> > > > > >
> > > > > > **Table 3.** Inference Time and Performance Comparison (OOD Cars)
> > > > > > | Base Model | Base Model | ActMAD | ActMAD (higher complexity) |$IT^{3}$|
> > > > > > |------------|------------|------------|---------|-|
> > > > > > | Inference Time | 1×         | 4x        |  7x    |5x|
> > > > > > | MAE ($\downarrow$) | 0.501         | 0.502        |  0.502    |0.424|
> > > > > >
> > > > > >
> > > > > > **Table 4.** Inference Time and Performance Comparison (Roads)
> > > > > > |                      | Base Model | Standard TTT | ActMAD  | ActMAD (higher complexity) | $IT^{3}$ |
> > > > > > |----------------------|------------|--------------|---------|----------------------------|--------|
> > > > > > | Inference Time       |     1x     |      3x      |  4.5x   |           8.5x             | 6x     |
> > > > > > | Quality  ($\uparrow$)|    39.5    |     40.0     |  45.9   |           46.3             |  69.8  |
> > > > > >
> > > > > > * **Comparison with TTA Methods**
> > > > > >
> > > > > > Thank you for raising the importance of TTA comparisons. We acknowledge that TTA and TTT have overlapping goals but operate under different assumptions. From one perspective, TTA can be seen as more constrained, as it assumes no control over the training process. On the other hand, TTT imposes its own form of strictness by resetting the model after every instance (outside the online variant) and treating each input as entirely independent.
> > > > > >
> > > > > > This independence in TTT creates a unique challenge: the method cannot retain information between test examples, even when they might share correlations or dependencies. In contrast, TTA benefits from access to data streams or batches during inference, allowing it to exploit temporal or structural relationships between inputs. IT$^3$’s online variant leverages such relationships, leading to higher performance in that setting, but we emphasize that it addresses a different scenario.
> > > > > >
> > > > > > To date, no prior TTT work has compared directly to TTA methods, largely because these paradigms are tailored to distinct challenges. However, we recognize the importance of acknowledging these distinctions and will add clarifications to this effect in the revised paper.
> > > > > >
> > > > > > * **"A Variety of Different Tasks" – IT$^3$’s Generality**
> > > > > >
> > > > > > We appreciate your observation of IT$^3$’s ability to operate across "a variety of different tasks." This reflects its **core strength**: IT$^3$ is a method designed to function across all modalities, architectures, and tasks without requiring task-specific adaptations.
> > > > > >
> > > > > > We do not claim that no domain-specific method could outperform IT$^3$ in specific tasks; indeed, highly tailored approaches may excel in certain settings (as noted in the limitations section in the paper). However, IT$^3$ offers unmatched **general applicability**, eliminating the need for task-specific tuning. This universal applicability sets IT$^3$ apart, functioning out of the box while maintaining robustness and consistency across diverse datasets and architectures.
> > > > > >
> > > > > > Among TTT works, ActMAD is indeed the most general approach. However, it relies on batch statistics, which limits its applicability in small or single-instance batch scenarios. Even within this constraint, we have directly compared to ActMAD wherever possible and achieved favorable results.
> > > > > >
> > > > > >
> > > > > > This generality is a fundamental distinction: IT$^3$ should not be evaluated as simply another method competing for top results in specific benchmarks but as a **general framework** for test-time adaptation across all modalities, architectures, and tasks.
> > > > > >
> > > > > > * **Paper Updates**
> > > > > >
> > > > > > We will incorporate the new results, along with all suggested revisions and additional clarifications, in the camera-ready version. Thank you for helping us improve the clarity and rigor of our work.

---

### Official Review · Reviewer_Tinn · 2024-11-03

**Soundness:** 2
**Presentation:** 2
**Contribution:** 2
**Rating:** 3
**Confidence:** 3

**Summary:**

This paper introduces $IT^3$, a generic method for addressing distribution shift challenges. By enforcing idempotence, $IT^3$ sequentially adapts to potential distribution shifts during inference. The authors show the performance of $IT^3$ across diverse scenarios.

**Strengths:**

The authors extend the concept of TTT (Sun et al., 2020) by incorporating idempotence, offering a simple yet elegant solution. The approach is intuitive and Figure 1 is particularly helpful for quickly grasping the core idea, even for those who are not experts in distribution shift.

**Weaknesses:**

1. (cf Figure 1) During training, in addition to feeding the standard $(x, y)$ pair to train the model, the authors also input the $(x, 0)$ pair to ensure the model satisfies the property of idempotence, referring to the zero as a neutral “don't know” signal. While this approach may work in classification tasks where zero could represent a new “don’t know” class, in regression tasks, it is unclear how the model differentiates this zero from an actual label of 0.
2. (Line 211) Online $IT^3$ appears to rely significantly on the use of Exponential Moving Average (EMA). However, the authors did not provide a citation for this technique.
3. In the first experiment (Section 4.1 on tabular data), the method of randomly substituting features with zeros in the test set may resemble a missing data scenario rather than a true distribution shift. In other experiments, the authors simulate distribution shift by splitting the training and testing data based on a specific covariate to create distinct distributions. It is unclear why the same method was not applied for the first experiment. If the authors prefer using the zeroing method, they should include figures or statistical tests to substantiate that the training and testing data are distributionally different, rather than relying solely on intuition.
4. There are unnecessary abbreviations throughout the manuscript. For instance, “such that” is shortened to “s.t.” in line 490. The proposed method, $IT^3$, is inconsistently referred to as ITTT in parts of the manuscript, such as in Figure 13.
5. Figures 14 to 16 are not mentioned or referred to in the main text. This omission is unusual and may confuse readers as to the purpose or relevance of these figures.
6. Figures 12 and 15 have the exact same title.

Some minor improvements and spelling corrections for clarity:
1. (Line 22) $x$ (not $\mathbf{x}$) is not mentioned before.
2. (Line 77) $y_2$ is not mentioned before.
3. (Line 85) "th input" should be corrected to "the input".
4. (Line 131) Missing right parenthesis: $f(f(z))$.
5. (Line 490) “s.t. that” should be replaced with “such that”.
6. (Line 495) Remove the extra colon (":").
7. In Table 1, the title references “qualitative” results, but the data presented are numerical and should be described as “quantitative” results.

**Questions:**

1. Do you use a special encoding or placeholder value instead of 0 for regression tasks?
2. If the purpose of the neural "don't know" zero is purely for contrast and its specific value is not important, will it be more computationally efficient to use a representative value such as the median of $y_i$ where $i \in \text{training set}$?
3. In EMA (Morales-Brotons et al., 2024), when $\alpha = 1$, the online $IT^3$ aligns with the offline version, and when $\alpha = 0$, it encounters the collapse issue described in Section 3.2. Could you provide guidance on selecting the value of $\alpha$ or share experimental results demonstrating performance across different $\alpha$ values?
4. The right panel of Figure 2 and Figure 16 both appear to represent the car data. Could there be a potential mistake or duplication here?

Morales-Brotons, D., Vogels, T., & Hendrikx, H. (2024). Exponential moving average of weights in deep learning: Dynamics and benefits. Transactions on Machine Learning Research.

---

> ### Author Response · Authors · 2024-11-23
> **Rebuttal by Authors**
>
> We sincerely thank the reviewer for their thoughtful feedback on our work. We appreciate their recognition of the simplicity and elegance of our approach, as well as its intuitive incorporation of idempotence into TTT. We value the reviewer’s insights and have carefully addressed all comments and concerns, as outlined below.
>
> ---
>
> * **In regression tasks, it is unclear how the model differentiates this zero from an actual label of 0.**
>
> We thank the reviewer for their important comment. The special 0 notation used does not represent an actual zero. Instead, it is a unique signal with the same dimensions as the labels, specifically chosen to differentiate it from actual label values. Our approach builds upon the ZigZag method from [1], where the choice of this signal is extensively discussed and justified. We understand that this notation may appear confusing, so we will revise the paper to include a clearer explanation to avoid any ambiguity.
>
> * **Online appears to rely significantly on the use of Exponential Moving Average (EMA). However, the authors did not provide a citation for this technique.**
>
> Thank you for pointing out. We will add citation for the mentioned paper [2]. It is indeed related and analyzes EMA. The origin of the technique is much older. We also want to point out that the reliance on EMA is not as strong as it may appear. The most important difference is- not resetting the weights after processing each input. Actually, in some cases we did set $\alpha=1$.
>
> * **In the first experiment (Section 4.1 on tabular data), the method of randomly substituting features with zeros in the test set may resemble a missing data scenario rather than a true distribution shift.**
>
> Thank you for making this distinction. Our goal in making this choice was to have a diversity of corruptions to showcase along the experiments. While acknowledging the distinction you make, we point that missing information makes a divergence from the training distribution, therefore can also be thought of as a distribution shift.
>
> * **There are unnecessary abbreviations throughout the manuscript. Figures 14 to 16 are not mentioned or referred to in the main text. Figures 12 and 15 have the exact same title. Some minor improvements and spelling corrections**
>
> Thank you for carefully reviewing and helping us improve our paper. In the revised paper these issues will be fixed.
>
> * **Do you use a special encoding or placeholder value instead of 0 for regression tasks?**
>
> Please see the explanation above: it is a special signal rather than an actual zero. For example, for tasks like aerodynamic predictions, we use placeholder values significantly different from the possible range of lift-to-drag and drag values to ensure clear differentiation.
>
> * **If the purpose of the neural "don't know" zero is purely for contrast and its specific value is not important, will it be more computationally efficient to use a representative value such as the median?**
>
> Thank you for the suggestion. However, the specific value is actually important. The model needs to clearly differentiate between the two inference modes: with and without additional information provided. For this reason, we use values that are significantly different from potential predictions. Additionally, this question pertains to the ZigZag [1] approach rather than the $IT^{3}$ method. We follow the guidelines outlined in the ZigZag paper, where this choice is extensively discussed.
>
> * **Could you provide guidance on selecting the value of or share experimental results demonstrating performance across different values?**
>
> Thank you for the suggestion. The ZigZag [1] paper provides detailed guidance on selecting placeholder values, emphasizing the importance of using values significantly different from valid predictions to distinguish inference modes effectively. Our experiments align with their findings, and we refer readers to their discussion for further details
>
> * **In EMA [2], when $\alpha=1$, the online IT$^3$ aligns with the offline version, and when $\alpha=0$, it encounters the collapse issue described in Section 3.2. Could you provide guidance on selecting the value of $\alpha$ or share experimental results demonstrating performance across different values?**
>
> The online version differs from the base version because they aim at different scenarios. The main difference is that in the base version the network weights are reset back to the state they were at the end of the pre-training phase. The assumption is that each input is a separate test and has no correlation or information about other inputs. The assumption for which the online version is made for, is correlation and somewhat smooth transitioning between inputs arriving in a stream. This is a continual learning regime. Therefore in this case, instead of resetting after each input, we leave the weights updated from the previous inputs. We have added a clarification to the paper.

---

> > ### Author Response · Authors · 2024-11-23
> >
> > * **The right panel of Figure 2 and Figure 16 both appear to represent the car data. Could there be a potential mistake or duplication here?**
> >
> > Thank you for pointing this out. There is indeed a labeling mistake in the right panel of Figure 2. It represents results for the UCI dataset, but it was incorrectly described as being for the cars dataset. We will correct this in the revised version.
> >
> > [1] Durasov, Nikita, et al. "Zigzag: Universal sampling-free uncertainty estimation through two-step inference." TMLR 2024.
> >
> > [2] Morales-Brotons, D., Vogels, T., & Hendrikx, H. (2024). Exponential moving average of weights in deep learning: Dynamics and benefits. TMLR 2024.

---

> > > ### Author Response · Authors · 2024-11-25
> > >
> > > Dear Reviewer Tinn,
> > >
> > > Thank you for dedicating your time to reviewing our paper and providing valuable feedback.
> > >
> > > We have thoughtfully addressed each of your comments, offering detailed responses to clarify and resolve the concerns you raised. We hope our explanations have provided a clearer perspective on our work and its contributions.
> > >
> > > If you feel that we have adequately addressed your concerns, we would be grateful if you would consider reassessing your rating.
> > >
> > > We would be happy to clarify or elaborate on any part of our paper while the discussion period is still open.
> > >
> > > Thank you!

---

### Official Review · Reviewer_hxXL · 2024-11-04

**Soundness:** 2
**Presentation:** 4
**Contribution:** 2
**Rating:** 3
**Confidence:** 3

**Summary:**

- This paper proposes a test-time-training based approach to address the distribution shift or OOD generalization problem by learning models that are idempotent.

- In particular, this paper proposes a method where models $f_{\theta}: \mathcal{X} \times \mathcal{Y} \to \mathcal{Y}$ are trained by minimizing both the difference between $f_{\theta}(x, y)$ and $y$ as well as the difference between $f_{\theta}(x, 0)$ and $y$, resulting in a model that is idempotent on the training set. Then at test time, models are optimized on the test data before running inference to make them idempotent on test inputs, by minimizing the difference between $f_{\theta}(x, 0)$ and $f_{\theta}(x, f_{\theta}(x, 0))$ for an OOD input $x$.

- The paper shows empirically for several different settings that idempotent test-time-training improves classification accuracy on out-of-distribution samples.

**Strengths:**

- Idempotent training is an interesting and novel approach for addressing the out-of-distribution generalization problem.

- The paper shows successful application of the considered approach on a large number of experimental settings, including in online-learning settings.

**Weaknesses:**

- Test-time training requires undesirable extra compute for test-time optimization of the whole model. How much additional compute is needed compared to running inference on the base model? How does this method scale with model size?

- The paper lacks analysis that explains how or why idempotent training is expected to improve out-of-distribution analysis. Further investigation and ablations that provide intuition for how this method works would be valuable.

- The proposed method has not been evaluated on standard real-world out-of-distribution generalization benchmarks such as DomainBed [Gulrajani and Lopez-Paz 2020] and WILDS [Koh et al 2020]. The presented experiments are on smaller models/datasets.

References:

Gulrajani, Ishaan, and David Lopez-Paz. "In search of lost domain generalization." arXiv preprint arXiv:2007.01434 (2020).

Koh, Pang Wei, et al. "Wilds: A benchmark of in-the-wild distribution shifts." International conference on machine learning. PMLR, 2021.

**Questions:**

- Does the test time objective result in networks that are idempotent on OOD samples? Presumably once the function drifts from the fixed anchor function that test time loss no longer reflects idempotence? It would be good to see measures of idempotence on training and test samples.

- How does the intuition about idempotence being a generalization of orthogonal projection hold up? The proposed method considers idempotence only in the y-variable, not the entire function.

---

> ### Author Response · Authors · 2024-11-23
> **Rebuttal by Authors**
>
> We sincerely thank the reviewer for their detailed and thoughtful feedback on our work. We deeply appreciate their recognition of the strengths of our approach, including the clarity of our presentation and the robustness of our experimental evaluation. We value their constructive suggestions and have carefully addressed all raised concerns, as outlined below.
>
> ---
>
> * **Test-time training requires undesirable extra compute for test-time optimization of the whole model. How much additional compute is needed compared to running inference on the base model? How does this method scale with model size?**
>
> We thank the reviewer for their question about computational costs. Test-time training (TTT) inherently induces additional computational overhead because optimization steps are performed during inference. In our method, while we require an additional inference step (two forward passes), this is computationally efficient since forward passes are generally 3–4x faster than backward passes. Thus, the extra inference step adds relatively little overhead compared to the total cost of TTT methods. We added a discussion about efficiency to the revised paper.
>
> Our method typically requires only 1–3 optimization steps, keeping the overall cost comparable to other well-known TTT methods. Below, we provide a comparison of inference times and performance improvements on out-of-distribution data for three approaches: the base model without optimization, the state-of-the-art TTT method ActMAD, and $IT^{3}$. As shown, while our method introduces no significant overhead compared to ActMAD, it delivers a substantial improvement in prediction accuracy (measured by MAE in Airfoils and Cars experiments and by Quality in Road experiment), whereas ActMAD achieves only marginal gains.
>
> **Table 1.** Inference Time and Performance Comparison (OOD Airfoils)
> | Base Model | Base Model | ActMAD | $IT^{3}$ |
> |------------|------------|------------|---------|
> | Inference Time ($\downarrow$) | 1×         | 3x        | 4x     |
> | MAE ($\downarrow$) | 38.67         | 38.61        | 37.5     |
>
> **Table 2.** Inference Time and Performance Comparison (OOD Cars)
> | Base Model | Base Model | ActMAD | $IT^{3}$ |
> |------------|------------|------------|---------|
> | Inference Time ($\downarrow$) | 1×         | 4x        | 5x     |
> | MAE ($\downarrow$) | 0.501         | 0.502        | 0.424     |
>
> **Table 3.** Inference Time and Performance Comparison (OOD Roads)
> | Base Model | Base Model | ActMAD | $IT^{3}$ |
> |------------|------------|------------|---------|
> | Inference Time ($\downarrow$) | 1x         |    4.5x     | 6x     |
> | Quality ($\uparrow$) | 39.5         | 45.9        | 69.8     |
>
>
> We will add this table to the revised version. We acknowledge that both our method and TTT methods in general introduce computational overhead during inference. This is a shared limitation across existing TTT approaches. Addressing this challenge and developing TTT methods that avoid such overhead entirely is an important research direction and could lead to significant advancements in the field.
>
> Regarding scalability with model size, our method works effectively with larger models, as demonstrated on aerial segmentation models with millions of parameters. By requiring only two forward passes and a small number of optimization steps, our approach remains computationally efficient even for such large-scale architectures, making it competitive with traditional TTT methods that often involve several backward passes.
>
> * **The paper lacks analysis that explains how or why idempotent training is expected to improve out-of-distribution analysis. Further investigation and ablations that provide intuition for how this method works would be valuable.**
>
> In the revised version, we will add a more concrete explanation about the reasons we expect idempotence to improve OOD performance. In the introduction lines 77-86 we build the intution towards the motivation to enforce idempotence. There are two prespectives to view this relation. The first one is that the idempotence term rises when minimizing the measure for OOD-ness from [1]. Then having idempotence as the objective makes sense according to [2] - The joint internal representation of (x,y) is projected onto the manifold of representations of (x,y) pairs that are in the data distribution. (More detailed explanation is in the repsponse to your last comment below).

---

> > ### Author Response · Authors · 2024-11-23
> >
> > * **The proposed method has not been evaluated on standard real-world out-of-distribution generalization benchmarks such as DomainBed [Gulrajani and Lopez-Paz 2020] and WILDS [Koh et al 2020]. The presented experiments are on smaller models/datasets.**
> >
> > We make a distinction between TTT and Test-Time-Adaptation. This distinction was also made by prior works eg., [3], [4]. In the TTT regime, there is no access to any data other than the instance/batch currently being processed. Furthermore, Each input is a separate test and has no correlation or information about other inputs. Prior works did not evaluate on the mentioned datasets and our model is demonstrated on various modalities, not just visual inputs. At the same time, our experiments on road segmentation (in terms of number of pixel) and aerodynamics with cars (in terms of number of nodes) can be considered large-scale, comparable in size to ImageNet.
> >
> > * **Does the test time objective result in networks that are idempotent on OOD samples? Presumably once the function drifts from the fixed anchor function that test time loss no longer reflects idempotence? It would be good to see measures of idempotence on training and test samples.**
> >
> > This is a valid point. The second application of the model does remain frozen which means that the objective diverges from idempotence. However, typically 1-3 steps are applied and this divergence is neglible. We have in fact experimented with having the second application an updated model (not with gradients, which would be wrong, but by copying the parameters from the updated model after each iteration as done in [2]). We found no significant difference in the performance.
> >
> > In [this plot](https://imgur.com/a/xuKEKXF), we show the distributions of idempotence loss for different subsets of the airfoils dataset: train, test, and OOD (both optimized and non-optimized versions). As seen, the train and test subsets exhibit lower idempotence loss values, while OOD samples have significantly larger losses. After optimization on OOD data, the idempotence loss values shift closer to those of the train and test subsets, supporting our point that the optimization helps align OOD samples with the in-distribution behavior.
> >
> > * **How does the intuition about idempotence being a generalization of orthogonal projection hold up? The proposed method considers idempotence only in the y-variable, not the entire function.**
> >
> > Idempotence is enforced over the y-variable given a specific x-variable. While $x$ itself is static throughout the TTT optimization, its representations by the network activations shifts as the network weights are updated. Internally the network has a joint representation for (x,y) at every layer, that can be considered as the input to the next layers. The objective of idempotence for y means that the model trains to make that internal representations such that they remain the same when applying the model again. So the representations of x,y are projected onto the manifold of the representations of x,y pairs that are in the data distribution. We will add this clarification to the paper.
> >
> > [1] Durasov, Nikita, et al. "Zigzag: Universal sampling-free uncertainty estimation through two-step inference." TMLR 2024.
> >
> > [2] Shocher, Assaf, et al. "Idempotent generative network." ICLR 2024.
> >
> > [3] Sun, Yu, et al. "Test-time training with self-supervision for generalization under distribution shifts." ICML 2020.
> >
> > [4] Gandelsman, Yossi, et al. "Test-time training with masked autoencoders." NeurIPS 2022.

---

> > > ### Author Response · Authors · 2024-11-25
> > >
> > > Dear Reviewer hxXL,
> > >
> > > Thank you for dedicating your time to reviewing our paper and providing valuable feedback.
> > >
> > > We have thoughtfully addressed each of your comments, offering detailed responses to clarify and resolve the concerns you raised. We hope our explanations have provided a clearer perspective on our work and its contributions.
> > >
> > > If you feel that we have adequately addressed your concerns, we would be grateful if you would consider reassessing your rating.
> > >
> > > We would be happy to clarify or elaborate on any part of our paper while the discussion period is still open.
> > >
> > > Thank you!

---

### Meta-Review · Area_Chair_TXXj · 2024-12-22

**Metareview:**

This paper addresses the test time training problem to handle distribution shifts. The paper proposes a method to learn a model that is idempotent. This is an interesting idea which the reviewers appreciated. The authors' rebuttal was considered and some of the reviewers also engaged in discussions with the authors.

However, after the discussions the reviewers still expressed several concerns, such as lack of analysis regarding why idempotent training is helpful for out of distribution settings as pointed out by Reviewer hxXL.

Reviewer ALWg raised some concerns about the extra inference time and the authors reported some additional experiments and promised to include these results in the revised manuscript.

Reviewer x3JD expressed concerns regarding lack of strong baselines and comparisons with TTA methods.

I appreciate the authors efforts to report some additional results in response to the reviewers' comments. However, the reviewers still had their reservations and, in the current state even with these results considered, the paper does not appear ready to be accepted. The paper's idea is interesting and incorporating the suggestions from the reviewers will definitely strengthen the paper. The authors are advised to address the issues and consider resubmitting to another venue.

**Additional Comments On Reviewer Discussion:**

Please refer to the meta-review.

---

### Decision · Program_Chairs · 2025-01-22

Reject